# Multi-modal investigation reveals pathogenic features of diverse *DDX3X* missense mutations

**Federica Mosti**[1,2], **Mariah L. Hoye**[1], **Carla F. Escobar-Tomlienovich**[1,2],
**Debra L. Silver**[1,2,3,4,*]

1 Department of Molecular Genetics and Microbiology, Duke University School of Medicine, Durham, North Carolina, United States of America, 2 Department of Neurobiology, Duke University School of Medicine, Durham, North Carolina, United States of America, 3 Department of Cell Biology, Duke University School of Medicine, Durham, North Carolina, United States of America, 4 Duke Institute for Brain Sciences and Duke Regeneration Center, Duke University School of Medicine, Durham, North Carolina, United States of America

* debra.silver@duke.edu

## Abstract

*De novo* mutations in the RNA binding protein DDX3X cause neurodevelopmental disorders including *DDX3X* syndrome and autism spectrum disorder. Amongst ~200 mutations identified to date, half are missense. While *DDX3X* loss of function is known to impair neural cell fate, how the landscape of missense mutations impacts neurodevelopment is almost entirely unknown. Here, we integrate transcriptomics, proteomics, and live imaging to demonstrate clinically diverse *DDX3X* missense mutations perturb neural development via distinct cellular and molecular mechanisms. Using mouse primary neural progenitors, we investigate four recurrently mutated *DDX3X* missense variants, spanning clinically severe (2) to mild (2). While clinically severe mutations impair neurogenesis, mild mutations have only a modest impact on cell fate. Moreover, expression of severe mutations leads to profound neuronal death. Using a proximity labeling screen in neural progenitors, we discover *DDX3X* missense variants have unique protein interactors. We observe notable overlap amongst severe mutations, suggesting common mechanisms underlying altered cell fate and survival. Transcriptomic analysis and subsequent cellular investigation highlights new pathways associated with *DDX3X* missense variants, including upregulated DNA Damage Response. Notably, clinically severe mutations exhibit excessive DNA damage in neurons, associated with increased cytoplasmic DNA:RNA hybrids and formation of stress granules. These findings highlight aberrant RNA metabolism and DNA damage in DDX3X-mediated neuronal cell death. In sum our findings reveal new mechanisms by which clinically distinct *DDX3X* missense mutations differentially impair neurodevelopment.

## Author summary

*DDX3X* mutations are associated with neurodevelopmental disorders, including *DDX3X* syndrome and autism spectrum disorder. DDX3X is an RNA binding protein whose

**Data availability statement:** All transcriptomic data in Fig 3 and S2 Table have been deposited in GEO with accession number GSE279078. All proteomics data (raw and normalized) are included in Fig 5 and S3 and S4 Tables. All numerical data for graphs and summary statistics underlying the findings are included in Supporting Information.

**Funding:** This work was supported by the following grants: National Institutes of Health (www.nih.gov) R01NS083897, R01NS120667, R37NS110388, R01MH132089, R21HD104514, and Ruth K. Broad Foundation (https://sites.duke.edu/broadfoundation/) to D.L.S.; National Institutes of Health F32NS112566 and Regeneration Next grant (https://sites.duke.edu/dukeregenerationcenter/) to M.L.H. M.L.H. is currently employed at the National Institutes of Health. This work was completed while M.L.H. was employed at Duke University. The opinions expressed in this article are the author's own and do not reflect the views of the National Institutes of Health, the Department of Health and Human Services, or the United States Government. Duke University Tricem Award to F.M. The funders had no role in study design, data collection and analysis, decision to publish, or preparation of the manuscript.

**Competing interests:** The authors have declared that no competing interests exist.

loss of function impairs brain development and generation of neurons. In this study, we focus on disease-associated missense mutations containing single amino acid changes in the *DDX3X* gene. We functionally probe two clinically severe mutations, associated with severe brain anatomical defects and intellectual disability, and two clinically mild mutations, characterized by modest anatomical disruptions and behavioral phenotypes. Using transcriptomics, proteomics, and live imaging, we identify distinct mechanisms by which clinically diverse *DDX3X* missense mutations disrupt neurodevelopment. Our findings reveal severe variants impair neuronal production and survival, by altering RNA metabolism and DNA damage response.

## Introduction

The cerebral cortex controls our abilities to process outside information and generate appropriate behavioral responses. The foundational basis for these complex and essential tasks are established during embryonic development. In the embryonic cortex, neurons and glia are generated by radial glial progenitors and basal progenitors [1–3]. Spatial and temporal regulation of progenitor behavior relies on precise coordination of gene expression. In particular, post-transcriptional regulation, including translation, is essential for proper brain development, and associated with diverse neurodevelopmental disorders [4–7].

Mutations in the RNA binding protein DDX3X cause diverse neurodevelopmental pathologies including autism spectrum disorder (ASD) and intellectual disability [8–10]. These neurodevelopmental outcomes are dubbed as *DDX3X* syndrome, which is also characterized by brain anatomical abnormalities, microcephaly, and associated behavioral, motor deficits, epilepsy, and language deficits [9,11–14]. *DDX3X* is located on the X chromosome, and the majority of affected individuals with *DDX3X* syndrome are females. Notably, a growing number of males have been identified suggesting that some *DDX3X* mutations may be tolerated in males [12,15,16]. Beyond neurodevelopment, missense mutations in *DDX3X* also cause diverse cancers including medulloblastoma [17–20].

*DDX3X* syndrome arises primarily from *de novo* mutations in *DDX3X*, with over 1000 reported cases internationally (ddx3x.org) [9,11–14]. About half of the approximately 200 cases for which molecular lesions are known are nonsense, whereas the remaining half are missense. *DDX3X* missense mutations are found preferentially in its helicase and RNA binding domains [9,12,14,21]. Based on clinical presentations and neuroimaging it has been hypothesized that there are two classes of missense mutations, clinically severe and mild. The former is defined by the presence of polymicrogyria (PMG, condition characterized by unusually small and dense brain gyri) and thinning or absence of the corpus callosum. In contrast the mild mutations generally resemble loss-of-function presentations, showing milder corpus callosum defects and absence of PMG. However, the molecular and cellular mechanisms by which these distinct *DDX3X* variants cause disease are largely unknown.

*DDX3X* encodes an ATP-dependent RNA helicase crucial for multiple aspects of RNA metabolism [22], including translation and RNA stability [23] . DDX3X is a nuclear-cytoplasmic protein, though primarily cytoplasmic in neural cells [9]. Consistent with this, it is especially implicated in translation of mRNAs with highly structured 5' untranslated regions (UTRs) [24]. Missense mutations in the helicase domains differentially affect DDX3X's RNA unwinding activity, with clinically severe and mild mutations showing either complete loss or hypomorphic activity, respectively [9,24,25]. DDX3X is thought to function within ribonucleoprotein (RNP) granules [26]. Notably, DDX3X undergoes liquid–liquid phase separation

(LLPS) and overexpression of *DDX3X* missense variants causes stress granule formation in cancer cells and neural cells [9,27–32]. Thus, stress granule formation is a measurable feature which may provide insight into diverse pathogenic mechanisms across *DDX3X* mutations.

Transient CRISPR and conditional knock-out (cKO) mouse models have demonstrated that *Ddx3x* loss-of-function (LoF) impairs cortical development [9,33,34]. Complete loss of *Ddx3x* in neural progenitors and their progeny causes profound microcephaly and apoptosis, whereas heterozygous females and hemizygous males have normal brain size but defective neurogenesis [34]. *Ddx3x* germline haploinsufficiency also causes behavioral and motor deficits postnatally [33]. These studies inform mechanisms of action for nonsense mutations and demonstrate essential roles of DDX3X in cortical development. However, a key question is how the landscape of *DDX3X* missense mutations affect neurodevelopment.

In this study, we focused on four recurrent *DDX3X* missense mutations spanning different clinical severity (DDX3X^R326H, DDX3X^R376C, DDX3X^R488H and DDX3X^T532M) (Fig 1A and 1B and S1 Table). We leverage cellular and molecular analyses using primary mouse neural cells to characterize differences in cell fate during early neurodevelopment, and discover changes in the transcriptome, protein interactome and subcellular localization across the *DDX3X* missense mutations. Our study demonstrates the differential impact of *DDX3X* missense mutations on neurodevelopment at cellular and molecular levels, furthering our understanding of the etiology of *DDX3X* syndrome.

## Results

### Expression of *DDX3X* missense mutations in mouse primary neural progenitors through lentiviral delivery

Individuals carrying *DDX3X* missense mutations present with a spectrum of clinical and molecular phenotypes, encompassing neuroanatomical disruption, developmental disability, and altered biochemical activity. To model *DDX3X* syndrome in primary neural cells, we focused on four *de novo* *DDX3X* missense mutations, which cause diverse clinical presentations and molecular features, and are recurrently found in 4–8 individuals each, all females. This includes DDX3X^R376C and DDX3X^R488H, which are clinically mild and retain some helicase activity, and DDX3X^R326H and DDX3X^T532M, which are clinically severe and helicase-dead (Fig 1B and S1 Table) [9]. MRI of patients with clinically mild mutations show corpus callosum defects and decreased white matter volume (data available for 5/8 DDX3X^R376C patients, 4/5 DDX3X^R488H patients) [9,12,14]. Severe mutations are associated with partial agenesis of corpus callosum and reduced white matter volume, accompanied by PMG and, in some cases, microcephaly and developmental delay (data available for 4/4 DDX3X^R326H patients, 4/5 DDX3X^T532M patients) (Fig 1B and S1 Table) [9,14,21].

A challenge in studying RBPs and missense variants, including DDX3X, is that overexpression can cause cellular phenotypes such as apoptosis [5,7,35,36]. Thus, we developed a lentivirus strategy to introduce *DDX3X* missense variants into mouse neural progenitors without overexpression. We isolated primary neural progenitor cells (NPCs) and neurons from embryonic day (E)12.5 mouse cortices, pooling brains from independent embryos to create a homogenous cell population, which was then used for primary cell culture. These cells were transduced with lentivirus containing GFP-tagged wild-type (WT) or mutant human (*Hs*) *DDX3X*. Importantly, mouse and human DDX3X show 98.6% identity at the protein level. After a two-day period for lentiviral expression, cells were analyzed (Fig 1C). We optimized the lentivirus amount and timing for these experiments by monitoring levels of induced GFP-tagged HsDDX3X protein by western analysis (Fig 1D–1F). We identified experimental conditions in which total DDX3X protein levels were relatively equivalent to WT and not

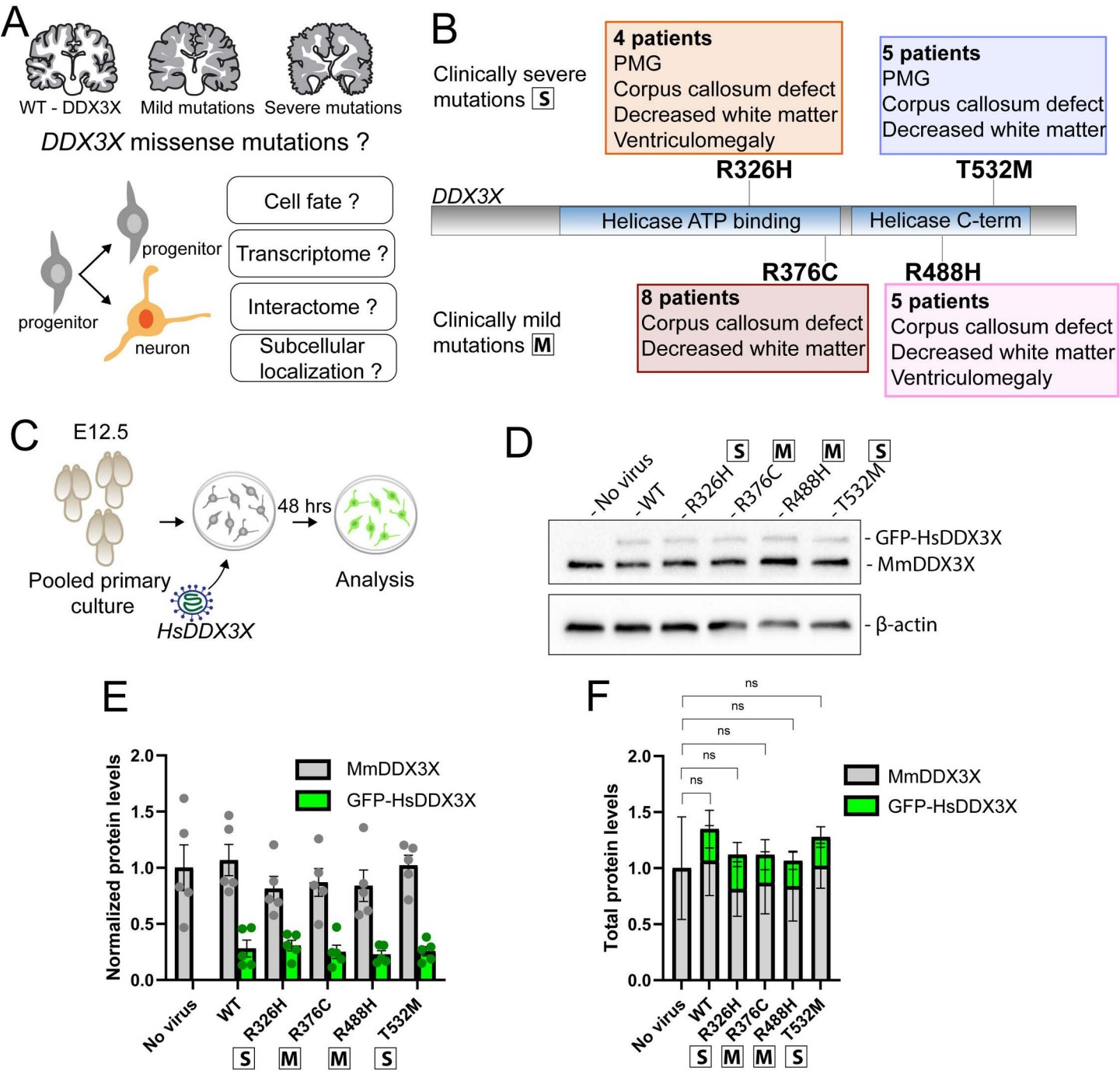

**Fig 1. Paradigm for expression of *DDX3X* missense mutations in primary mouse neural progenitors.** A) (Top) Cartoon of MRIs from WT, mild and severe *DDX3X* missense mutations; (bottom) representation of dividing neural progenitor (grey) generating a progenitor and a neuron (orange), and overview of the questions assessed in this study. B) Schematic of DDX3X with helicase domains indicated. The 4 missense mutations analyzed in this study and their main brain malformations are annotated. C) Schematic of experimental paradigm. Mouse brains were dissected at E12.5, dissociated and prepared as primary cell cultures from pooled brains. *HsDDX3X* was introduced through lentiviral delivery. After 48 hrs incubation, the cells were harvested or fixed for analysis. D) Western blot of mouse primary cells 2 days after lentiviral delivery of GFP-*HsDDX3X*. E) Quantification of D. F) Total DDX3X protein levels in transduced cells: statistical analysis was performed on total amount of protein (endogenous + exogenous). Protein levels were normalized to β-actin protein levels. Each dot represents a pooled primary culture. (F) Two-way ANOVA, Ns, not significant. Data are mean ± SD. "S": severe; "M": mild.

over-expressed, with the missense variants expressed at about 30% that of the endogenous protein (Fig 1E and 1F). Further, using fluorescence microscopy, we demonstrated DDX3X was transduced in virtually all cells (S1A and S1B Fig). Amongst transduced cells, there was some variation in GFP levels. However, this occurred equivalently across all *DDX3X* variants. These experiments establish an efficient strategy for evaluating the phenotypic impact of *DDX3X* mutations upon neural cells.

### *DDX3X* missense mutations differentially impair neuron production and apoptosis

Previous studies showed that *Ddx3x* LoF disrupts progenitor cell cycle and its ability to produce neurons, however whether missense mutations also affect progenitors is unknown [34]. Using our paradigm for expression of *DDX3X* missense variants, we next assessed their functional impact upon cell fate. Towards this, we utilized a mouse line expressing *Dcx*::DsRed as a reporter for newborn neurons [37]. Lentiviral delivery of *DDX3X* was performed as in Fig 1C, followed by live imaging for 24 hours, and assessment of progeny, as previously [38] (Fig 2A). We first assessed the extent to which *DDX3X* missense mutations impact progenitors' ability to divide. Progenitors expressing DDX3X$^{R376C}$, DDX3X$^{R488H}$, and DDX3X$^{T532M}$, each exhibited successful divisions 100% of the time. In contrast, DDX3X$^{R326H}$ expression led 20% of progenitors to fail cell division (S1C and S1D Fig). Mitosis duration was not overtly different across the variants (S1E Fig). However, DDX3X$^{T532M}$ and DDX3X$^{R488H}$ displayed a higher rate of progenitor re-divisions (S1F and S1G Fig). This suggests that these mutations may affect progenitor cell cycle.

We next assessed if *DDX3X* missense mutations impact the type of divisions which progenitors undergo. Proliferative symmetric divisions were defined by production of two DsRed− cells, neurogenic asymmetric divisions by one DsRed− (progenitor) and one DsRed+ cell (neuron), and neurogenic symmetric divisions by two DsRed+ cells (Fig 2B). Progenitors expressing the clinically severe *DDX3X* mutations (DDX3X$^{R326H}$ and DDX3X$^{T532M}$) exhibited significantly more proliferative divisions and fewer symmetric neurogenic divisions, relative to control cells (Fig 2C). In comparison, expression of clinically mild variants had only subtle impacts on cell fate, with no alterations observed with DDX3X$^{R488H}$ and only slightly fewer asymmetric divisions with DDX3X$^{R376C}$ (Fig 2C). Strikingly, all *DDX3X* missense mutations induced apoptosis in progeny which we identified by cell morphology as previously [38] (Fig 2D and 2E). These data demonstrate that *DDX3X* missense mutations differentially impact progenitors' ability to produce viable neurons, with clinically severe mutations causing significant impairment of neurogenesis.

We also assessed cells that did not divide during the live imaging session to measure the extent to which *DDX3X* mutations influence survival of either progenitors (dsRed−) or neurons (dsRed+). Amongst all four mutations, only DDX3X$^{R376C}$ expression led to a slight but significant increase in progenitor death compared to DDX3X$^{WT}$ (Fig 2F). However, in neurons, expression of either clinically severe mutation, DDX3X$^{R326H}$ and DDX3X$^{T532M}$, led to a striking threefold increase in apoptosis as compared to DDX3X$^{WT}$ and clinically mild mutations (Fig 2G). These analyses indicate that *DDX3X* missense mutations influence cell viability, with a particularly profound impact of the clinically severe mutations upon neuronal survival.

### *DDX3X* clinically severe missense mutations induce diverse transcriptomic changes in neural progenitors and neurons

We next investigated the underlying molecular mechanisms by which clinically distinct *DDX3X* missense mutations influence cell fate. Towards this, we conducted transcriptome

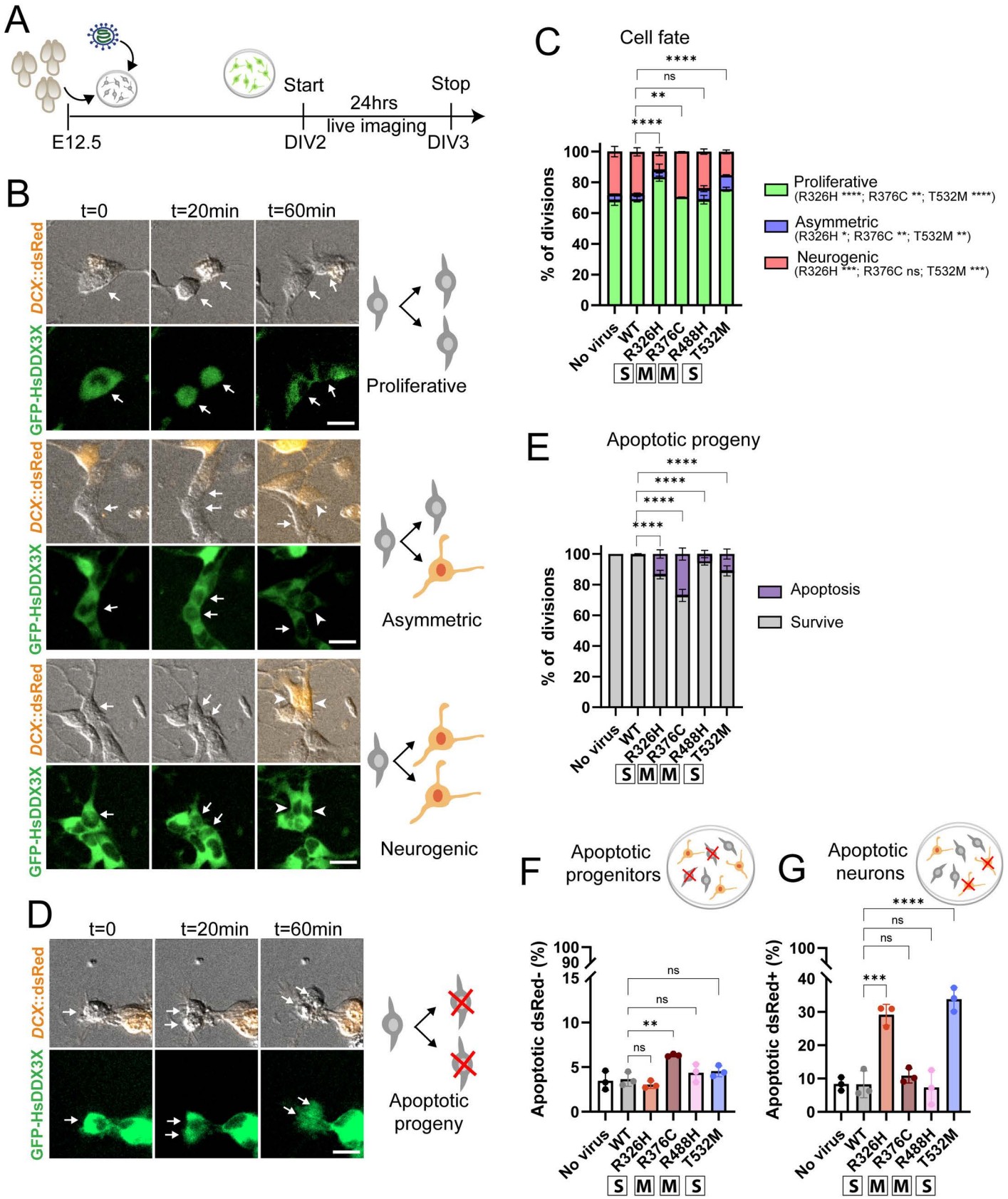

**Fig 2. *DDX3X* missense mutations impair neural cell fate and survival.** A) Schematic of the experimental timeline. Neural progenitor primary cells were isolated at E12.5 and DDX3X$^{WT}$ and missense mutations were introduced through lentiviral delivery. After 2 days *in vitro* (2 DIV), live imaging was performed for 24hrs.

B) Example of proliferative, asymmetric and neurogenic divisions. Arrows, neural progenitors (DCX::dsRed−): arrowheads, newborn neurons (DCX::dsRed+). C) Quantification of cell fate for proliferative symmetric (green), asymmetric (blue) and neurogenic asymmetric (red) divisions. D) Example of division with apoptotic progeny. E) Quantification of apoptotic progeny. F) Quantification of the percentage of progenitors that underwent apoptosis over total number of dsRed− cells. G) Quantification of the proportion of neuron that underwent apoptosis over total number of dsRed+ cells. (C, E) $\chi^2$ analysis with post-hoc Bonferroni. n = 3 live-imaging sessions, n = 3 litters; no virus n = 545 divisions, $DDX3X^{WT}$ n = 645 divisions, $DDX3X^{R326H}$ = 365 divisions, $DDX3X^{R376C}$ n = 410 divisions, $DDX3X^{R488H}$ n = 649 divisions, $DDX3X^{T532M}$ n = 651 divisions. Scale bars: 10 μm. (F, G) Each dot represents a pooled primary culture. Two-way ANOVA. **p < 0.001; ****$p <$ 0.0001; ns, not significant. Data are mean±SD. "S": severe; "M": mild.

analysis of pooled primary cells either without lentiviral transduction (control) or with lentiviral-mediated expression of DDX3X$^{WT}$ or missense mutations (Fig 3B). Importantly, DDX3X$^{WT}$ expressing cells did not display any differentially expressed transcripts compared to control (Fig 3D and S2 Table). Notably, in all conditions (WT and 4 mutations), there were no transcriptional changes in *Ddx3x* (total exogenous and endogenous) (S2A and S2C Fig). These data further confirm that our experimental setup did not induce *Ddx3x* overexpression.

We then examined the extent to which the global transcriptome was affected by *DDX3X* mutations (S2 Table). We identified extensive differentially expressed transcripts in cells expressing the severe mutations (DDX3X$^{R326H}$: 331 DDX3X$^{T532M}$: 259) (Fig 3A, 3C, 3E and 3H). In contrast, despite expression at similar levels, the mild mutations impacted far fewer transcripts (DDX3X$^{R376C}$: 94; DDX3X$^{R488H}$: 74 (Fig 3A, 3C, 3F and 3G). Notably, most of these differentially expressed transcripts were also present in the severe mutation datasets (Fig 3A and 3C). The *DDX3X* paralog on the Y chromosome, *Ddx3y*, was upregulated 3-fold in all missense mutation datasets relative to WT (S2B Fig). Notably, the number of reads mapped on the X and Y chromosome was consistent between experimental replicates (S2C Fig). Upregulation of *Ddx3y* has been previously observed in *Ddx3x* LoF brains, suggesting that it compensates for reduced *Ddx3x* levels in this model [34]. The upregulation of *Ddx3y* suggests it also could compensate for possible *DDX3X* sub-optimal function associated with the missense mutations [39]. Together, these transcriptome data reveal divergent impacts of missense mutations on the cellular transcriptome, with the most profound disruption evidenced with severe mutations.

Gene Ontology (GO) analysis of differentially regulated transcripts across all the *DDX3X* missense mutations revealed enrichment in categories associated with cellular death, cellular stress, P53 activation and DNA damage (Figs 3A and S2D and S2 Table). The transcripts associated with DNA damage were all upregulated, including several kinases involved in P53 activation (*Atm*, *Atr*, *DNA-PK*, and *Parp1*) [40] (Fig 3A, and 3E–3H). We independently validated the increased expression of *Atm* and *Parp1* using qPCR of cells expressing *DDX3X* missense mutations (Fig 3I and 3J). These transcriptome findings highlight altered pathways associated with DNA damage response, including some which may be due to defective DNA replication. This is notable given our finding that *DDX3X* missense mutations caused extensive apoptosis (Fig 2).

### *DDX3X* clinically severe missense mutations induce DNA damage and cause aberrant R-loop formation

Given these transcriptomic changes, we postulated that *DDX3X* mutations may impair genomic stability. To test this, we first assessed if expression of *DDX3X* missense variants activates the DNA damage response (DDR) gene, P53. For this we quantified the fraction of GFP+P53+ cells. All missense mutations led to P53 activation in more than 80% of cells, while control and DDX3X$^{WT}$ did not (Fig 4A and 4D). Similarly, all missense mutations exhibited a striking increase in pH2A.X+ cells (DNA double strand breaks), with the clinically severe

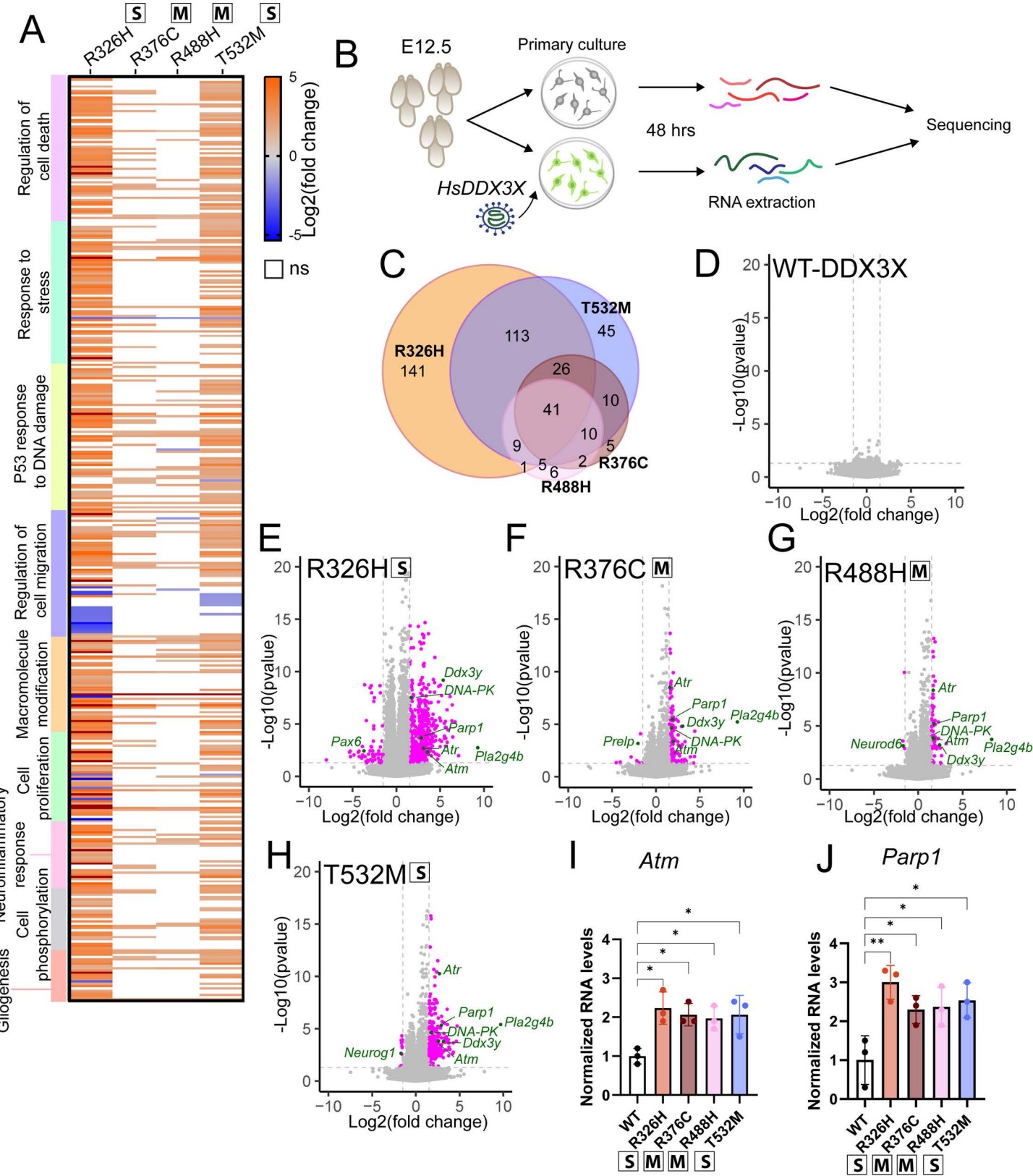

**Fig 3. Transcriptome analysis reveals *DDX3X* missense mutations alter expression of DNA damage response genes.** A) Heatmap of differentially expressed (upregulated: orange, downregulated: blue, not significant: white) transcripts across the four missense mutations, organized by Gene Ontology categories. B) Schematic of

experimental design: RNA was extracted from primary cultures with and without viruses and used for bulk RNA sequencing. C) Venn diagram of significant differentially regulated transcripts across all four mutations relative to control. D, E, F, G, H) Volcano plots of transcriptome datasets for DDX3X$^{WT}$ and each indicated missense mutation. Each dot represents a transcript. Pink: transcript fold change >1.5 or <−1.5, p-value <0.05; grey: not significant; green: selected hits of biological significance. I, J) RNA levels of *Atm* (I) and *Parp1* (J) measured by qPCR. RNA levels normalized of *Gapdh*. Each dot represents pooled primary culture. Two-way ANOVA. *$p < 0.01$; **$p < 0.001$. Data are mean ± SD. "S": severe; "M": mild.

mutations (DDX3X$^{R326H}$ and DDX3X$^{T532M}$) showing damage in ~60% of cells (Fig 4B and 4E). These data are consistent with the transcriptomic signature of elevated DDR in cells expressing all mutations, but especially severe variants. DNA damage was highest in Tuj1+ neurons, especially in those expressing clinically severe mutations (Fig 4B and 4F). This demonstrates that neurons expressing severe mutations are especially prone to DNA breaks, consistent with the higher apoptosis of neurons seen with expression of these specific variants (Fig 2G).

As DDX3X is an RNA binding protein, we postulated that *DDX3X* mutations may induce DNA damage by influencing RNA metabolism. Of note, DDX3X binds not only RNA:RNA duplex but also ssDNA and DNA:RNA hybrids (R-loops) *in vitro* [41,42]. R-loops are essential for DNA replication and RNA transcription, and defective resolution of these structures has been associated with increased DNA damage [43]. Thus, we hypothesized that *DDX3X* missense mutations may increase R-loops in neural cells, causing cytoplasmic accumulation as has been recently reported in immortalized cells [43–46]. DDX3X$^{R376C}$ and DDX3X$^{R488H}$ mutations did not affect the presence of R-loops (S9.6+) (Figs 4C and 4G and S2F). This is consistent with the hypomorphic impact of these mutations upon RNA helicase activity [9] as well as relatively mild induction of DNA damage (Fig 4E). In contrast, cells expressing either clinically severe mutation showed significant cytoplasmic accumulation of R-loops (Fig 4C and 4G). Low levels of S9.6 were detectable in the nucleus, supporting the notion that this signal is due to co-transcriptional R-loop formation (S2E Fig) [43]. Together, our findings reveal that expression of *DDX3X* missense mutations, but not DDX3X$^{WT}$, induces DNA damage and P53 activation. Clinically severe mutations cause especially high DNA damage, particularly in neurons, and this is associated with defective RNA metabolism.

## *DDX3X* missense mutations have both shared and unique protein interactomes

To further understand how missense mutations impact RNA metabolism and cellular fate, we next examined their protein interactomes. We thus conducted proximity labeling to discover the protein interactome of DDX3X$^{WT}$ and missense mutations within neural cells in an unbiased fashion. For this we used TurboID fused to either DDX3X$^{WT}$ or missense protein variants. TurboID, a biotin ligase that biotinylates nearby proteins, is commonly used to investigate proteins in close proximity and presumed interactors [47] (Fig 5A). These were expressed in primary neural progenitors using lentivirus as in Fig 1. Biotinylated proteins were then affinity purified using streptavidin beads and subjected to quantitative mass spectrometry. Samples were prepared for TurboID alone (negative control), DDX3X$^{WT}$, and mutations shown in Fig 1B. Unique peptides identified for each sample were normalized to the negative control, resulting in more than 300 significantly enriched proteins for each condition (fold change > 1.5; p-value <0.05; S3 and S4 Tables).

Proximity labeling revealed both unique and shared protein interactomes across the different *DDX3X* variants. We identified 211 protein interactors shared between DDX3X$^{WT}$ and all missense mutations (Figs 5B and 5C and S3A and S4 Table). Notably, these shared protein partners were primarily enriched for mRNA metabolism categories, including mRNA splicing

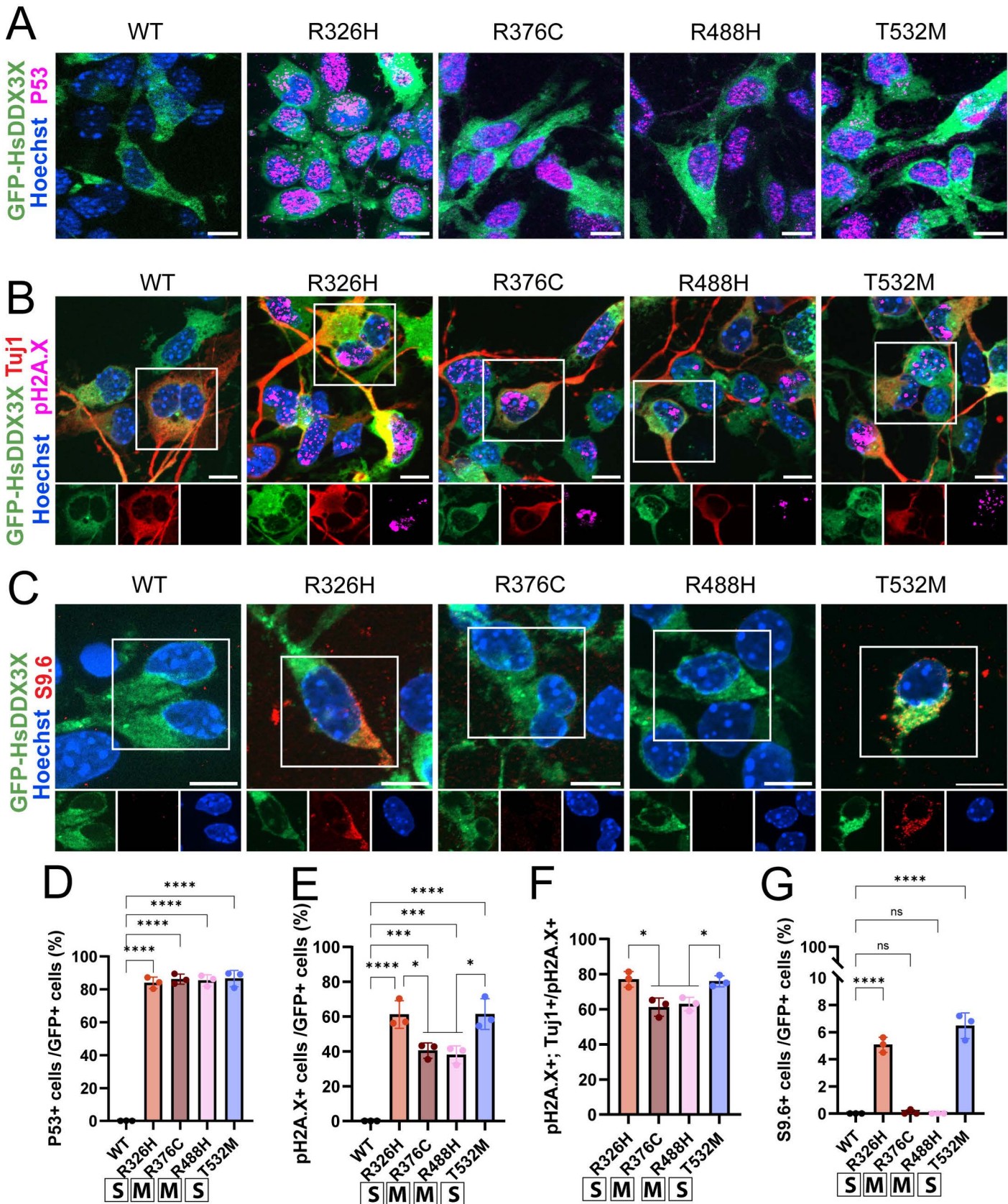

**Fig 4. Clinically severe *DDX3X* missense mutations induce DNA damage in neurons and accumulation of R-loops in the cytoplasm.** A) Representative images of P53 staining (magenta), GFP-HsDDX3X (WT or indicated mutation, green), and Hoechst (blue) in primary cells. B) Representative staining of Tuj1 (neurons,

red) and pH2A.X (double strand breaks, magenta), GFP-HsDDX3X (WT or indicated mutation, green), and Hoechst (blue) in primary cultures. C) Representative images of S9.6 (R-loops, red), GFP-hsDDX3X (WT or indicated mutation, green), and Hoechst (blue) staining in primary cells. D) Quantification of the fraction of GFP+P53+ cells showed in A. E, F) Quantification of the fraction of GFP+pH2A.X+ (E) the fraction of GFP+pH2A.X+TuJ1+ (F) cells shown in B. G) Quantification of the fraction of GFP+S9.6+ cells shown in C. Scale bars: 10 μm. Two-way ANOVA *$p < 0.01$; **$p < 0.001$; ***$p < 0.0001$; ****$p < 0.0001$; ns, not significant. Eat dot represents a pooled primary culture. Data are mean ± SD. "S": severe; "M": mild.

(GO:0000389) and RNA processing (GO: 0006396). This demonstrates that *DDX3X* variants lacking helicase activity (DDX3X$^{R326H}$ and DDX3X$^{T532M}$) retain the ability to interact with RNA metabolism machinery (S3A–S3E Fig). Importantly, we also discovered DDX3X interactors previously observed in other cell types and species [26,30,48–54] (S3A Fig). This reinforces the robustness and validity of our datasets. Amongst these interactors was DDX3X itself, with similar fold enrichment across all mutations. This suggests that all mutations preserve the ability to homodimerize (Fig 5B).

We next compared only *DDX3X* missense variant interactomes to each other. Amongst these, the clinically severe mutations DDX3X$^{R326H}$ and DDX3X$^{T532M}$, showed the highest percentage of unique interactors (around 15–16% compared to 4–8% in the mild mutations) (Fig 5D). Strikingly these mutants had 40 protein interactors in common (Fig 5B and 5D). Gene ontology analysis showed that both DDX3X$^{R326H}$ and DDX3X$^{T532M}$ but not DDX3X$^{WT}$, interacted with proteins associated with noncoding post-transcriptional gene silencing (GO: 0016441) and regulation of protein localization (GO:0070201) (S4A–S4D Fig). While DDX3X$^{R488H}$ did not present enough uniquely enriched peptides to perform gene ontology analysis, DDX3X$^{R376C}$ displayed preferential interaction with proteins involved in DNA homeostasis (S4A–S4D Fig). The composition of the proximity labeling datasets suggests that while all *DDX3X* missense mutants interact with RNA metabolism proteins, clinically severe mutations acquire new mutation-specific interactors.

To validate the DDX3X protein interactors, we focused on novel DDX3X interactors not previously reported in other cell types. We prioritized interactors with the highest enrichment amongst DDX3X$^{WT}$ and the four *DDX3X* missense variants (Fig 5B–5D). DDX3X$^{WT}$-specific interactor CFDP1 (Craniofacial Development Protein 1) is associated with cell cycle progression and the neurodevelopmental disorder Williams-Beuren syndrome [55]. Notably, the most enriched interactors of the missense mutants are associated with DNA damage (SMURF2 and RHOB, respectively DDX3X$^{R326H}$ and DDX3X$^{T532M}$ interactors) [56,57], stress granules and protein localization (HOOK2 and HSPBP1, respectively DDX3X$^{R376C}$ and DDX3X$^{R488H}$ interactors) [58–61]. These highlighted candidates were used to evaluate DDX3X variant-specific interactions in N2A cells.

We first validated interactions by co-expression of Flag-HsDDX3X and mScarlet-tagged interactors in N2A cells and performed co-immunoprecipitation (IP) using the Flag tag (Fig 5E–5N). To test for specificity, for each unique interaction we tested both Flag-DDX3X$^{WT}$ and the missense mutations (Fig 5E–5N). DDX3X$^{WT}$, but not DDX3X$^{R326H}$, pulled down CFDP1 (Fig 5E and 5F). Similarly, DDX3X$^{R326H}$, DDX3X$^{R376C}$, DDX3X$^{R488H}$, and DDX3X$^{T532M}$ mutant proteins each co-precipitated with their respective interactor, but not with WT protein (Fig 5G–5N). These data provide orthogonal validation of *DDX3X* variants' unique physical interactions.

We also validated the TurboID results by measuring sub-cellular co-localization in N2A cells co-expressing the GFP-HsDDX3X variant with its mScarlet-tagged interactor. Consistent with previous studies, DDX3X was primarily cytoplasmic with some DDX3X+ punctae in the nucleus (S5C Fig) [9]. In cells expressing DDX3X$^{R326H}$, CFDP1 was primarily nuclear showing

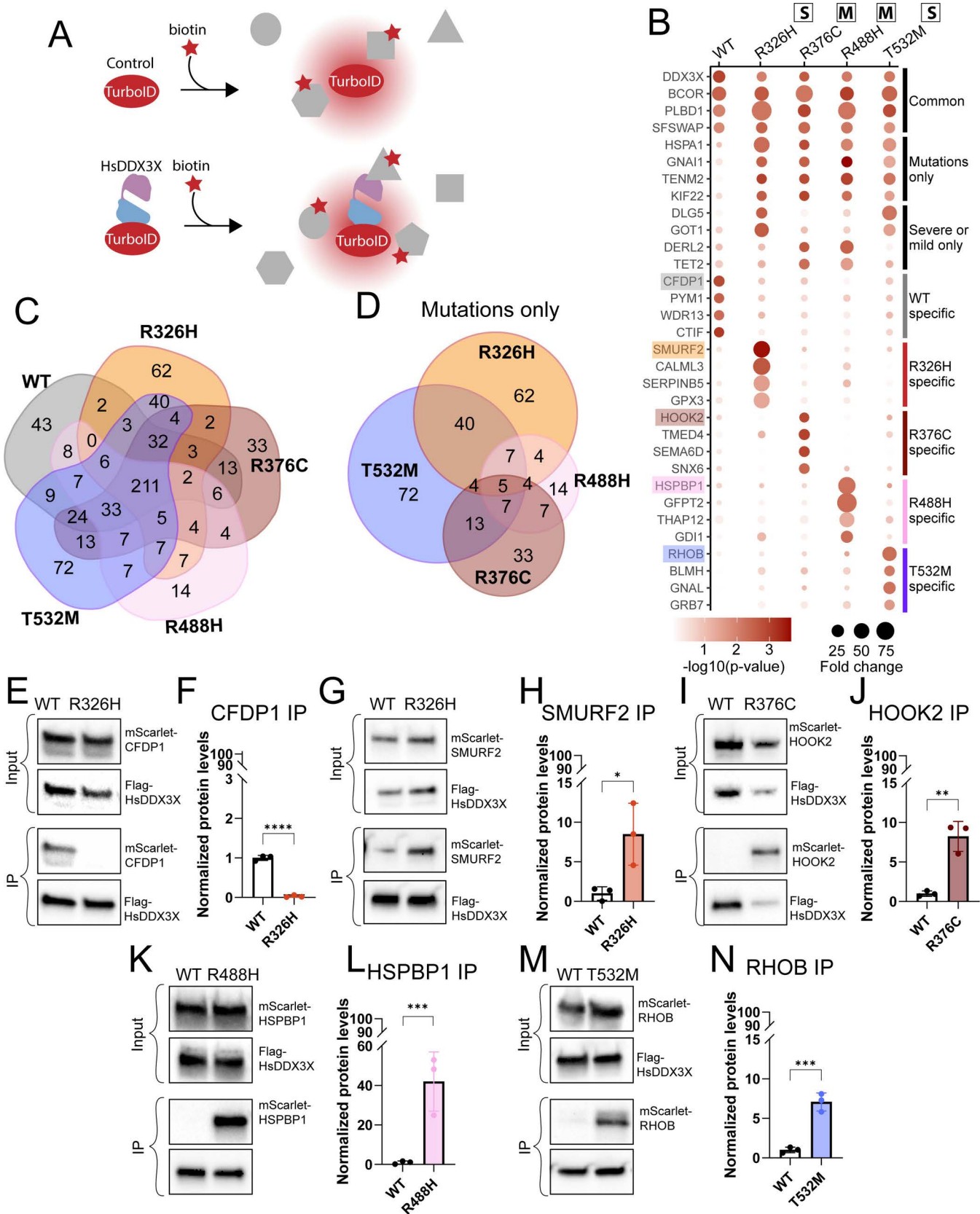

**Fig 5. Turbo-ID proximity labeling analysis reveals *DDX3X* mutations exhibit differential protein interactomes.** A) Schematic of experimental design. Negative control (unfused TurboID) or HsDDX3X-TurboID (WT or missense mutations) were delivered in primary cells. B) Scatter plot of top hits for

each sub-category (common, mutations only, severe or mild only, WT-specific, R326H-specific, R376C-specific, R488H-specific, T532M-specific). C) Venn diagram of significant interactors across all conditions. D) Venn diagram of all the significant interactors across the missense mutations and not shared with the WT condition. E-N) Western blots of immunoprecipitations (IPs) for targets highlighted in (B) performed in N2A cells and quantifications. Unpaired two-tailed t-test. *$p < 0.01$; **$p < 0.001$; ***$p < 0.0001$; ****$p < 0.0001$. Data are mean ± SD. "S": severe; "M": mild.

little overlap with DDX3X. In comparison, in the presence of DDX3X$^{WT}$, CFDP1 subcellular distribution was more cytoplasmic (S4E and S4F Fig). Co-expression of DDX3X$^{R326H}$ and SMURF2 resulted in cytoplasmic sequestration of the latter, a phenotype previously observed for SMURF2 in cancer cells [62] (S4G and S4H Fig). Under pathogenic conditions, HOOK2 is involved in formation of aggresomes, a pericentrosomal accumulation of misfolded proteins [63]. Consistent with this, DDX3X$^{R376C}$ and HOOK2 co-expression induced the formation of aggresomes [64] (S4I and S4J Fig). Moreover, while RHOB localized to the cell membrane in WT-expressing cells, co-expression with DDX3X$^{T532M}$ led to a higher percentage of cells with cytoplasmic localization (S4K and S4L Fig). In contrast, DDX3X$^{R488H}$ did not affect the localization of its top interactor (S4M and S4N Fig). Taken together, these data demonstrate that overexpression of *DDX3X* variants in neuronal cells can impact subcellular localization of protein interactors. Together this rich dataset highlights new binding partners which may inform the functional impact of *DDX3X* clinically severe and mild mutations.

## *DDX3X* missense mutations generate stress granules with specific cellular and physical characteristics

These localization studies indicate that *DDX3X* missense mutations can impact the subcellular distribution of DDX3X and their protein interactors. Notably, DDX3X is a component of cytoplasmic stress granules, and overexpression of *DDX3X* missense variants can induce cytoplasmic granules in neural and immortalized cells in the absence of stress [9,30]. This is relevant, given our finding that *DDX3X* missense mutations induce DNA damage (Fig 3), a well characterized source of cell stress [65]. Thus, we hypothesized that *DDX3X* missense mutations drive formation of stress granules in primary neural cells and this may be most prominent for those mutations associated with significant DDR. To test this, we used our lentivirus paradigm to introduce GFP-tagged *DDX3X* variants into primary neural cells. In the absence of external cell stress, we visualized localization of these variants (Figs 6A and S5B). Notably, DDX3X$^{WT}$ induced spontaneous granule formation in less than 1% of transduced cells, with an average of 1 granule per cell (Fig 6B and 6C). The clinically severe mutations (DDX3X$^{R326H}$ and DDX3X$^{T532M}$) showed granules in around 10% of transduced cells with a mean of 20 granules per cell. In contrast, the clinically mild mutations (DDX3X$^{R376C}$ and DDX3X$^{R488H}$) showed granules in only about 5% of cells, with an average of less than 10 granules per cell (Fig 6B and 6C). Additionally, the clinically severe mutations formed granules preferentially in neurons compared to WT, while DDX3X$^{R376C}$ showed the opposite trend, with more granules in progenitors (Figs 6D and 6E and S5A). These results demonstrate that *DDX3X* missense mutations have different propensity for granule formation, with the clinically severe variants exhibiting higher potency of formation, especially in neurons.

We next characterized the nature of these granules, as cytoplasmic granules induced with extrinsic stressors have distinct protein composition depending on cell types and disease analyzed [26]. Towards this, we characterized DDX3X granules by immunostaining for RNA binding proteins which can be recruited to stress granules (FMRP and G3BP1) [9,66]. Notably, almost a third of DDX3X$^{R326H}$ and DDX3X$^{T532M}$ granules colocalized with FMRP while the percentage was almost double for the mild mutations (Fig 7A and 7C). In contrast, neither

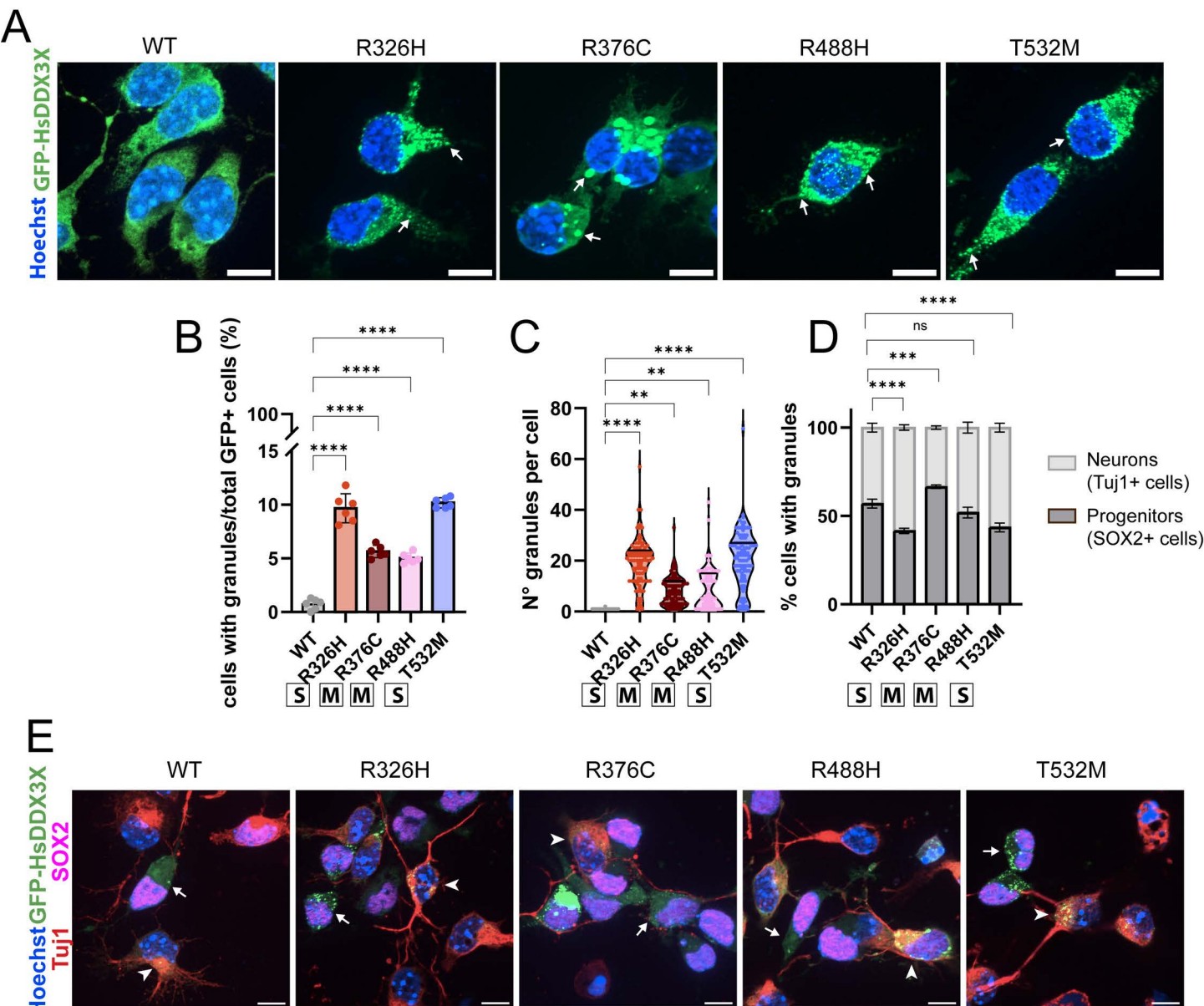

**Fig 6. *DDX3X* missense mutations cause formation of stress granules in primary neural progenitors and neurons.** A) Examples of GFP-HsDDX3X (green) sub-cellular distribution and stress granule formation in primary cells. Arrows point to examples of individual granules. B) Quantification of percentage of GFP+ cells with granules (n = 150–400 cells across 6 litters). C) Quantification of the average number of granules per GFP+ cell (n = 40–100 cells across 6 litters) D) Quantification of percentage of Tuj1+ or SOX2+ cells with DDX3X granules (N= 74–272 cells across 6 litters) E) Example of Tuj1 (red), SOX2 (magenta), GFP (green) and Hoechst (blue) staining in primary cultures. Arrows indicated SOX2+ cells with granules, arrowheads point to Tuj1+ cells with granules. Scale bars: 10 μm. Two-way ANOVA **$p < 0.001$; ***$p < 0.0001$; ****$p < 0.0001$; ns, not significant. Data are mean ± SD. "S": severe; "M": mild.

DDX3X$^{WT}$ nor any of the missense mutations overlapped with G3BP1 (Fig 7B and 7D). These observations are consistent with overexpression of DDX3X variants in N2A cells [9].

To investigate the physical properties of these granules, we characterized DDX3X granule dynamics using Fluorescence Recovery After Photobleaching (FRAP) (Fig 7E). In this assay we did not observe clear segregation of granule dynamics by clinical severity. Instead, the mutations in different protein regions tended to show similar properties. Mutations in

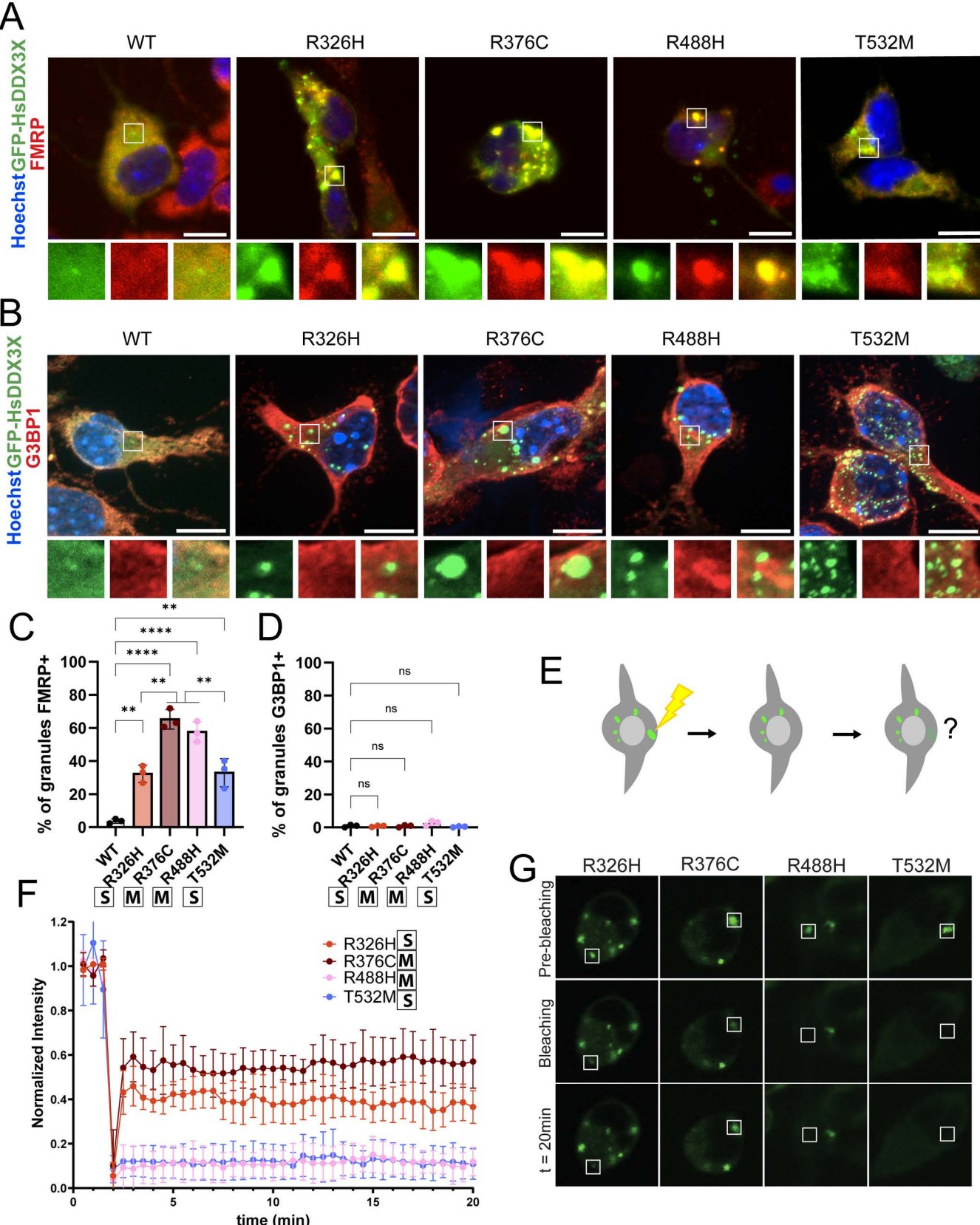

**Fig 7.** ***DDX3X* missense granules show differential protein composition and physical properties.** A) FMRP (red), GFP (green), Hoechst (blue) staining in primary cells. Inserts show overlap of FMRP and DDX3X in individual granules. B) G3BP1 (red), GFP (green), and Hoechst (blue) staining in primary

cells. Inserts show overlap of FMRP and DDX3X in single granules. C) Quantification of FMRP+ DDX3X granules. D) Quantification of G3BP1+ DDX3X granules. E) Schematic depiction of FRAP experiments. Single DDX3X granules were photobleached using a laser and recovery of fluorescence was followed for 20 min. F) Quantification of fluorescence intensity of DDX3X granules during FRAP assay. G) Examples of GFP-HsDDX3X signal before, during and after photobleaching. Scale bars: 10 µm. Two-way ANOVA. $*p < 0.01$; $**p < 0.001$; $***p < 0.0001$; $****p < 0.0001$; ns, not significant. Data are mean ± SD. "S": severe; "M": mild.

the N-terminal helicase domain (DDX3X$^{R326H}$ and DDX3X$^{R376C}$) showed a partial recovery of granule fluorescence after photobleaching. These showed differential dynamics, with DDX3X$^{R376C}$ exhibiting a ~60% recovery, and DDX3X$^{R326H}$ showing only a ~40% recovery (Fig 7F and 7G). This indicates that while both N-terminal mutations display limited granule dynamics, DDX3X$^{R326H}$ has a lower capability for cytoplasmic exchange. Mutations in the C-terminal DDX3X helicase domain (DDX3X$^{R488H}$ and DDX3X$^{T532M}$), did not show any fluorescence recovery after photobleaching of their granules (Fig 7F and 7G), demonstrating that granules formed by these specific mutations are more solid-like. These results show that *DDX3X* C-terminal missense mutations induced granules which exhibit reduced molecular exchange with the cytoplasm. The differences observed in the dynamics and composition of these granules thus might contribute to the variable clinical severity associated with *DDX3X* mutations.

## Discussion

*DDX3X* has emerged as a central causal gene for neurodevelopmental pathologies including ASD and *DDX3X* syndrome. Yet, we lack a fundamental understanding of how clinically diverse mutations impact DDX3X molecular and cellular functions. Here, we employ a multi-modal investigation of cell fate, subcellular localization, binding partners and molecular targets to discover common and unique mechanisms of *DDX3X* syndrome (Fig 8). We demonstrate for the first time that missense mutations impair neural development and discover distinct molecular signatures associated with aberrant neurogenesis and neuronal survival. We discover that clinically severe mutations perturb neuronal fate by influencing DNA damage and RNA metabolism. Our work nominates new cellular and molecular mechanisms by which *DDX3X* mutations impair brain development, shedding light on their potential impact on neuronal health and disease progression.

The impact of *DDX3X* upon neurodevelopment is influenced by its known ability to escape X inactivation as well as presence of the DDX3Y paralog [67]. The extent of X inactivation escape in the developing brain is unknown, but could impact mosaicism in the brain and/or altered DDX3X dosage [68,69]. Similar to LoF in mice [34], expression of *DDX3X* missense variants leads to upregulation of *Ddx3y*. While *Ddx3y* is posited to mitigate phenotypic impact in *Ddx3x* LoF males, *Ddx3y* upregulation following expression of missense mutants did not prevent severe phenotypes such as apoptosis. Our protein interaction studies indicate that the *DDX3X* missense variants retain interactions with the RNA metabolism machinery, suggesting they may potentially sequester DDX3Y protein partners, limiting a possible functional rescue. Thus, it is important to consider the complex interplay between DDX3X partners and redundancy with DDX3Y.

Clinically mild and severe *DDX3X* missense mutations impair neural cell fate and survival to different degrees. Our live imaging paradigm allowed us to assess the consequences of missense variants upon individual neural progenitors and neurons. Clinically severe variants led to a striking reduction in direct production of neurons by progenitors. Further, these variants result in significant apoptosis, evident in about 30% of all neurons. In comparison, mild

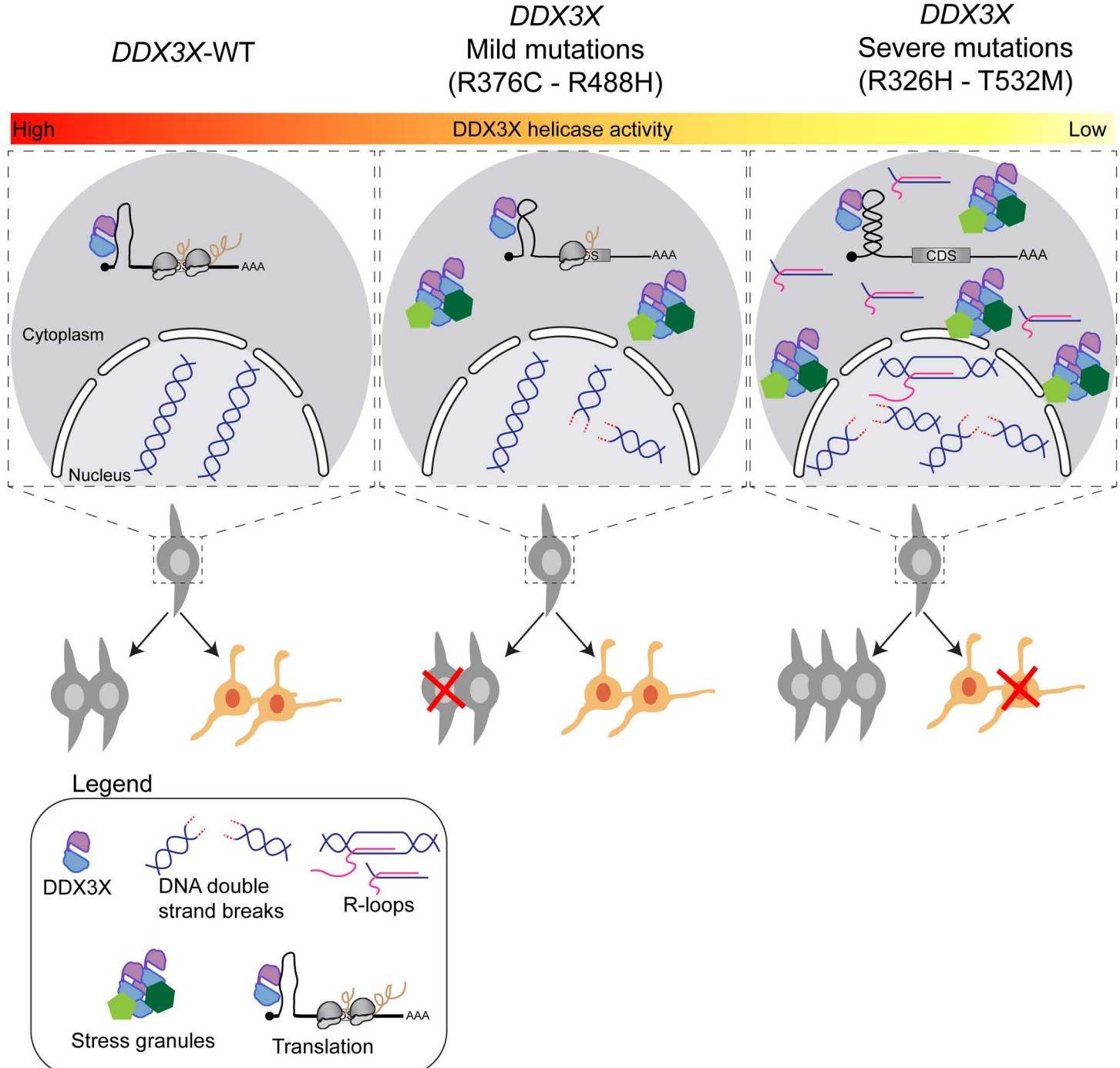

**Fig 8. Model depicting how mild and severe *DDX3X* missense mutations influence RNA metabolism, cell fate and survival.** Effects of DDX3X$^{WT}$ (left), clinically mild (middle) and severe missense mutations (right) upon RNA metabolism and neural cell fate. Clinically mild mutations show moderate effects on DDX3X helicase activity and lead to modest accumulation of DNA damage and stress granules, resulting in apoptosis in progenitors. Clinically severe mutations cause increased accumulation of DNA damage and translocation of aberrant R-loops into the cytoplasm, leading to formation of stress granules, aberrant neurogenesis and neuronal apoptosis.

mutations had subtle impacts on both neurogenesis and survival. It is notable and somewhat surprising that *Ddx3x* severe mutations impaired neurogenesis to a similar degree as LoF [34]. However, in contrast to the missense mutations, *Ddx3x* LoF in progenitors does not impact

progeny survival. It is possible that the acute introduction of missense variants explains this phenotypic difference, as LoF was genetically induced several days prior to analysis [34].

How do these developmental alterations contribute to clinical outcomes? Severe mutations are defined by the presence of PMG as well as thinning or absence of the corpus callosum [9,70,71]. The altered neurogenesis and increased neuronal death caused by expression of the severe variants suggest potential cellular defects which may contribute to these cortical malformations. In sum, taken with previous LoF studies [34], our data reinforce that functional *DDX3X* is necessary both for generation and survival of neurons.

We discover underlying molecular mechanisms to explain how *DDX3X* severe missense mutations impair cell survival. Our data indicate cell death is associated with excessive DNA damage and aberrant DNA:RNA hybrids. Notably, these identical severe *DDX3X* variants also profoundly reduce RNA helicase activity [9]. Thus, we postulate that *DDX3X* severe mutations cause unresolved R-loops as an early manifestation of impaired RNA helicase activity. In line with this model, extensive literature shows that unresolved R-loops cause DNA double strand breaks and activation of DDR [45,72,73]. Unresolved DNA damage and sustained DDR in neurons can induce their death. One mechanism for this is cell cycle reentry and activation of checkpoint genes such as ATM [74]. Thus, in neurons expressing *DDX3X* variants, upregulation of ATM/ATR signaling may cause DDR and apoptosis.

Our study establishes the first connection between R-loop formation and *DDX3X* missense mutations in the context of neurodevelopmental syndrome. Notably, downregulation of *DDX3X* in human cancer lines also causes R-loops [75]. Together this suggests that both loss of function and missense variants can interfere with resolving DNA:RNA hybrids.

Recent studies indicate R-loops can cause neuroinflammation [44,45,72]. Cytoplasmic R-loops can be detected by immune system receptors, resulting in activation of IRF3 and triggering innate immune response and apoptosis [45,46]. This raises the intriguing possibility that expression of severe *DDX3X* mutations may also be associated with unique immune responses. In future studies it will be interesting to assess neuro-immune outcomes associated with *DDX3X* syndrome, as well as how anti-inflammatory drugs such as IRF3 specific inhibitors [76] modulate *DDX3X* phenotypes.

Expression of clinically severe mutations also led to striking increases in stress granule formation. We postulate that this outcome results from a stressed cellular state associated with aberrant DNA damage and altered RNA metabolism. It is notable that DDX3X granules, like FUS granules, lack co-localization with G3BP1 [77,78]. While stress granules can form without G3BP1, its presence is necessary for granule disassembly [35,79,80]. Absence of this pivotal component hints at an inability of the cell to disassemble DDX3X stress granules leading to formation of chronically stable cytoplasmic aggregates. Differences in granule composition and stability between severe and mild mutations may reflect a pathological role of these organelles and/or a stressed cellular state. Future studies exploring the causal relationship between DDX3X granule formation, DNA damage, and neuronal viability will be critical for understanding their pathophysiological relevance in *DDX3X* syndrome.

Our study highlights transcriptomic and signaling pathways unique to each mutation suggesting possible biomarkers and future therapeutic strategies for *DDX3X* syndrome. To date, there is no approved therapy for *DDX3X* syndrome nor ongoing clinical trials. Identification and classification of specific molecular signatures associated with different *DDX3X* mutations might help to prioritize potential therapeutic strategies, including patient-specific efficacy and personalized therapies. Similar strategies are being applied for other developmental diseases [81–83] The clinical complexity and genetic heterogeneity of *DDX3X* syndrome generate challenges for therapeutic avenues, however continued molecular phenotyping of *DDX3X* variants is a valuable path.

Our study establishes a technical paradigm for a broader investigation of the diverse landscape of *DDX3X* missense mutations, as well as missense mutations in other genes linked to other neurodevelopmental disorders. Mutations in dozens of RNA binding proteins have been associated with neurodevelopmental disorders and intellectual disabilities [5,84]. A key challenge in studying DDX3X and other RNA binding proteins is that overexpression can cause cellular phenotypes such as apoptosis. Our lentiviral paradigm circumvents this challenge by enabling screening of disease variants with low-levels of expression. This strategy may be extended to more complex systems, such as human brain organoids, providing the opportunity to work directly in human models and without lengthy and expensive genome editing.

Similarly, our study highlights conceptual frameworks relevant for related neurological disorders. Observations that R loop formation occurs with missense variants reinforces the central role that aberrant RNA metabolism plays in neurodevelopmental pathologies. Chronic accumulation of R-loops is associated with both neurodegenerative diseases [43,44] and neurodevelopmental diseases [85]. Further, cytoplasmic inclusions are linked to neurodegenerative diseases [86], and our work further supports their association with neurodevelopmental pathology [36,87,88].

In conclusion, through comprehensive analyses of four recurrent *DDX3X* mutations we have uncovered new cellular and molecular mechanisms that explain how these mutations differentially impair neurodevelopment. By linking DNA damage, R-loop cytoplasmic accumulation, and stress granule formation to severe mutations, we offer insights into the pathophysiology of *DDX3X* syndrome. Taken together, these highlight new avenues of investigation into how specific DDX3X missense variants uniquely affect cell survival pathways via RNA metabolism and DNA damage. This work establishes a technical and conceptual foundation for future studies, enabling deeper exploration into the diverse landscape of DDX3X mutations and their contribution to neurodevelopmental disorders. Future research should focus on categorizing additional missense mutations to fully delineate the spectrum of *DDX3X*-related pathologies.

## Materials and methods

### Ethics statement

All animal procedures were approved by the Duke Institutional Animal Care and Use Committee (IACUC) and performed in agreement with the ethical guidelines of the Division of Laboratory Animal Resources (DLAR) from Duke University. We used the previously described mouse line: Dcx::DsRed [37]. The following mouse strains were obtained from Charles Rivers: CD1 (strain 022). For embryo staging, plug dates were defined as embryonic day (E) 0.5 on the morning the plug was identified.

### Primary cultures

Primary cortical cultures were derived from E12.5 embryonic dorsal cortices, as previously described in [38]. In brief, tissue was prepared in a single cell suspension by trypsinization (7 min with 0.25% Trypsin-EDTA, Thermo Fisher Scientific, 25200056), and mechanical dissociation by pipetting p200 pipette. 12–16 (from 6–8 embryos) cortices were pooled together and isolated cells were plated in neural progenitor media (DMEM Thermo Fisher Scientific 11965092; B-27 supplement Life Technologies 12587010; N2 supplement Thermo Fisher Scientific 17502048; N-acetyl cystein Sigma A9165; bFGF R&D 3139-FbB-025/CF). Cells were plated 24-well glass-bottom dish (MatTek P24G-1.5-10-F) and given 3h to settle at 37°C in 5% $CO_2$ prior to lentiviral transduction. After 48hrs from lentiviral transduction, the cells were harvested for analysis.

## Live imaging

Dcx::DsRed E12.5 cortices were used to derive primary culture as described above. 48hrs after viral transduction, the neural progenitor media was change. Images were captured every 20 min for 24 h using a 20× magnification on a Zeiss Axio Observer Z.1 equipped with a XL multi S1 incubation chamber, $CO_2$ module S, temperature module S, and humidity control. The cells were kept 37°C and 5% $CO_2$ for the duration of the live imaging. Mitosis duration, catastrophe and viability were identified by morphology (rounding of cells and visual identification of condensation of chromatin). Cells were considered neurons if expressing dsRed for at least 4 hours.

## Plasmid constructs

All cloning PCR products were amplified by Q5 polymerase (NEB, M049L). For construction of vectors used for lentiviral assays, HsDDX3X$^{WT}$ and missense mutations were amplified from previously generated plasmid [9] and inserted into the pUbc-EGFP (Addgene # 98916), EGFP or TurboID, by EcoRV enzyme sites. For generating mScarlet constructs, gene of interest were amplified from mouse cDNA and inserted in pCAGGS-mScarlet by BsrI enzyme site. HiFi DNA Assembly Master Mix (NEB, E2621L) was used to aligned cloned inserted and cut plasmids. Sequence was validated by Sanger sequencing.

## Lentivirus packaging and transduction

HEK293T cultured in DMEM+10%FBS+1%P/S were used for lentivirus production. 26.75ug pMF58 containing EGFP-HsDDX3X or TurboID-HsDDX3X (DDX3X$^{WT}$ or missense mutations), 20 µg Packaging plasmid (psPAX2) and 6.25 µg Envelope plasmid (pMD2.G) were transfect with PEI-MAX (Polysciences, 24765-100) when cells reached to 70–80% in 15 cm dish. 48hrs after transfection, the medium was collected and centrifuged at 5000 g × 10 min. The supernatant was then filtered through a 0.45 mm filter (VWR, 28143-352) into an ultracentrifuge tube (Beckman-Coulter, 357448). 4ml sterile 20% sucrose was added below the medium using a 5 ml serological pipette. The mixture was centrifuge at 19700 rpm for 2hrs at 4°C in a swinging bucket rotor (Beckman-Coulter, SW28). The supernatant was then removed, 100 µl of PBS was added, and the virus was resuspended overnight at 4°C with rocking. The resuspended virus was aliquoted, flash-freeze and stored at −80°C. To transduce cells, each lentivirus was titrated to achieve ~30% overexpression and added to the cells in the evening. After 16–18 hours, the media was changed. Nearly all cells were transduced by the following day (S1A and S1B Fig).

## Immunofluorescence staining

Cells were fixed in 4% PFA for 10min at room temperature. Permeabilization was done with 1X PBS/0.25% TritonX-100 for 10min and blocked with 5% NGS/PBS for 30min at room temperature. Cells were incubated with primary antibodies overnight at 4°C, and secondary antibodies at room temperature for 1 hours (Alexa Fluor-conjugated, Thermo Fisher, 1:800). The following primary antibodies were used: DDX3X (Sigma, HPA001648, 1:500), SOX2 (Thermo Fisher, 14-9811-82, 1:1000), FMRP (Sigma, F4055, 1:200), G3BP1 (Proteintech, 13057-2-AP, 1:100), TUJ1 (Biolegend, 801202, 1:1000), pH2A.X (Cell Signaling, 9718, 1:200), P53 (Leica, CM5, 1:250), S9.6 (Kerafast, ENH001, 1:100). Slides were mounted with Vectashield (Vector Labs, H-1000-10).

## Microscope image acquisition and processing

Images were captured using a Zeiss Axio Observer Z.1 equipped with an Apotome for optical sectioning at 20x (0.8 NA) (S1 Fig), 40x (1.4 NA) (Figs 4A and 4B, and 6E) and/or 63x (1.4

NA) (Fig 4C), alternatively Zeiss LSM 780 confocal microscope using 63x magnification (1.4 NA) (Figs 6A, 7A and 7B, S4E, S4G, S4I, S4K and S4M). Images were acquired using Zen Blue 2.6 (Zeiss Axio Observer Z.1) or Zen Black 2011 (Zeiss LSM 780). 10 fields of view were captured per experiment and all images for a giver experiment were captured with identical exposures. All images were captured as 16 bit and processed using ImageJ. Cells were manually counted (Fiji cell counter).

## RNA extraction and RT-qPCR

Cells were centrifuged to make them precipitate and RNA was purified from Trizol (Thermo, 15596026) method. cDNA was prepared using iScript kit (Bio-Rad, 1708891) following manufacturer's protocol. qPCR was performed using SYBR Green iTaq (Bio-Rad, 1725124). In at least three independent biological samples in a QuantStudio 3 machine (Applied System). Values were normalized to *Gapdh* as loading control. The following primers were used: *Mus musculus Gapdh* (Forward 5'-TGGATTTGGACGCATTGGTC-3' and Reverse 5'-TTTGCACTGGTACGTGTTGAT-3'), *Mus musculus Atm* (Forward 5'-GCTTCCTCCCGAAATTCCTGT-3' and Reverse 5'-CCTCTAAAGGGTCCCATTCGT-3') and *Mus Musculus Parp1* (Forward 5'-CTCTGTACTTTGAAAACCACCGT-3' and Reverse 5'-GCTCAGTCGGACACCATGTA-3').

## Bulk RNA sequencing and analysis

Primary culture cells were precipitated and RNA was purified RNAeasy kit (Qiagen), each experimental condition has 3 biological replicates. cDNA libraries were prepared by Illumina TruSeq stranded mRNA kit. RNAseq libraries were sequenced on the NovaSeq (PE100) with 20M paired end reads. RNAseq libraries were sequenced to a depth of ~40 million total reads per sample. RNASeq data aligned by Star Salmon from FASTAq files, and normalized on "no virus" samples. Differential gene expression (DGE) analysis using DESeq2. DGE lists were defined using an FDR <0.05, with Log2(FC) ≥ 1.5 or ≤ −1.5. Overlapped and specific gene expression changes were defined by Biovinn ([https://www.biovenn.nl/](https://www.biovenn.nl/)). GO analysis were performed on PANTHER [89] and categories were selected by FDR <0.05.

## Proximity labeling with TurboID and mass spectrometry

Primary culture cells from E12.5 embryos from 3 CD1 mouse litters (per trial to obtain 1 biological replicate for each lentivirus) were dissected, mixed, and plated such that each condition had the same background primary cells prior to transduction. Cells were transduced with TurboID or TurboID-DDX3X (DDX3X$^{WT}$ or missense mutations) for 48 hrs. 500uM biotin (Sigma) were added to the medium and incubate 45min at 37°C. Medium was removed and cells were washed with ice cold 1xDPBS 5 times on ice. Then, cells were lysate in 1x RIPA buffer (50 mM Tris-HCl pH = 7.4, 150 mM NaCl, 1% sodium deoxycholate, 2% Triton-X, 0.2% SDS) with protease inhibitors (Roche, 4693132001). Lysates were frozen at −80°C for 3min to further lyse the cells. Lysates were thawed and triturated 5 times with 25G needle, then, spun down at 15000 g for 10min at 4°C. The supernatants were frozen at −80°C. After three biological replicates for each condition were obtained, the supernatants were thawed and incubated with 50 µl of Streptavidin beads (Thermo Fisher, 65601) overnight at 4°C with end-over-end rotation. Beads were magnetized and washed with the following: 2× RIPA buffer, 2× 1M NaCl, 1× 2M urea, 2× 50 mM ammonium bicarbonate. Beads were eluted in 100 µl of elution buffer (25 mM tris, 50 mM NaCl, 10 mM DTT, 2% SDS, 5 mM free biotin) and boiled for 10 min at 98°C. 15 uL of input, supernatant, wash 1, and elution were saved to perform western blotting to confirm the IPs were successful. Elution was sent for mass spectroscopy at

the Duke Proteomics and Metabolomics Core Facility. Samples were spiked with undigested bovine casein at a total of either 1 or 2 pmol as an internal quality control standard. Next, samples were reduced (10 mM dithiolthreitol already in samples) for 30 min at 80C, alkylated with 20 mM iodoacetamide for 30 min at room temperature, then supplemented with a final concentration of 1.2% phosphoric acid and 866 µL (depending on sample volume) of S-Trap (Protifi) binding buffer (90% MeOH/100mM TEAB). Proteins were trapped on the S-Trap micro cartridge, digested using 20 ng/µL sequencing grade trypsin (Promega) for 1 hr at 47C, and eluted using 50 mM TEAB, followed by 0.2% FA, and lastly using 50% ACN/0.2% FA. All samples were then lyophilized to dryness. Samples were resolubilized using 12 µL of 1% TFA/2% ACN with 12.5 fmol/µL yeast ADH.

Quantitative LC/MS/MS was performed on 3 µL (25% of total sample) using an MClass UPLC system (Waters Corp) coupled to a Thermo Orbitrap Fusion Lumos high resolution accurate mass tandem mass spectrometer (Thermo) equipped with a FAIMSPro device via a nanoelectrospray ionization source. Briefly, the sample was first trapped on a Symmetry C18 20 mm × 180 µm trapping column (5 µl/min at 99.9/0.1 v/v water/acetonitrile), after which the analytical separation was performed using a 1.8 µm Acquity HSS T3 C18 75 µm × 250 mm column (Waters Corp.) with a 90-min linear gradient of 5–30% acetonitrile with 0.1% formic acid at a flow rate of 400 nanoliters/minute (nL/min) with a column temperature of 55C. Data collection on the Fusion Lumos mass spectrometer was performed for three difference compensation voltages (−40v, −60v, −80v). Within each CV, a data-dependent acquisition (DDA) mode of acquisition with a r = 120,000 (@ m/z 200) full MS scan from m/z 375–1500 with a target AGC value of 4e5 ions was performed. MS/MS scans were acquired in the ion trap in Rapid mode with a target AGC value of 1e4 and max fill time of 35 ms. The total cycle time for each CV was 0.66s, with total cycle times of 2 sec between like full MS scans. A 20s dynamic exclusion was employed to increase depth of coverage. The total analysis cycle time for each injection was approximately 2 hours.

Following UPLC-MS/MS analyses, data were imported into Proteome Discoverer 2.5 (Thermo Scientific Inc.). In addition to quantitative signal extraction, the MS/MS data was searched against the SwissProt *M. musculus* database (downloaded in Nov 2019), the human DDX3X sequence, and a common contaminant/spiked protein database (bovine albumin, bovine casein, yeast ADH, etc.), and an equal number of reversed-sequence "decoys" for false discovery rate determination. Sequest (v 2.5, Thermo PD) was utilized to produce fragment ion spectra and to perform the database searches. Database search parameters included fixed modification on Cys (carbamidomethyl) and variable modification on Met (oxidation). Search tolerances were 2ppm precursor and 0.8Da product ion with full trypsin enzyme rules. Peptide Validator and Protein FDR Validator nodes in Proteome Discoverer were used to annotate the data at a maximum 1% protein false discovery rate based on q-value calculations. Note that peptide homology was addressed using razor rules in which a peptide matched to multiple different proteins was exclusively assigned to the protein has more identified peptides. Protein homology was addressed by grouping proteins that had the same set of peptides to account for their identification. A master protein within a group was assigned based on % coverage. Prior to imputation, a filter was applied such that a peptide was removed if it was not measured in at least 2 unique samples (50% of a single group). After that filter, any data missing values were imputed using the following rules; 1) if only one single signal was missing within the group of three, an average of the other two values was used or 2) if two out of three signals were missing within the group of three, a randomized intensity within the bottom 2% of the detectable signals was used. To summarize to the protein level, all peptides belonging to the same protein were summed into a single intensity. These protein levels were then subjected to a normalization in which the top

and bottom 10 percent of the signals were excluded and the average of the remaining values was used to normalize across all samples. Each of the identified putative interactors of the DDX3X-fused conditions resulting values across the three biological values have been normalized over the negative control condition (TurboID). The normalized values have been used to calculate fold change and p-values (t-test). The significative hits (fold change >1.5, p-value <0.05) of the DDX3X$^{R326H}$, DDX3X$^{R376C}$, DDX3X$^{R488H}$ and DDX3X$^{T532M}$ have been also used to do normalization over DDX3X$^{WT}$ to analyzed changes between WT protein and missense variations.

## Western blot and co-immune precipitations

Cell cultures were lysed at 4°C in TNE buffer (50 mM Tris-HCl pH = 7.4, 137 mM NaCl, 0.1 mM EDTA) containing 0.5% Triton X-100, plus protease inhibitors (Roche). Protein lysates were incubated on ice for 10 min and clarified by centrifugation at 20,000 rcf for 5 min. For co-immunoprecipitations 1% octyl-beta-glucoside was added to the lysis buffer, and samples were incubated for 40 min on ice and clarified by manual spin for 15 sec at 4°C. 0.5–1 ml of each protein lysate was incubated with 50 µl anti-Flag M2 Magnetic beads (Sigma, M8823) overnight at 4°C with shaking. Beads were magnetized and after 5 washes with TNE buffer plus 0.5% NP-40 detergent and protein inhibitors, beads were resuspended in elution buffer (TNE buffer plus 1x laemmli buffer and 50 mM DTT) and boiled 5–10 min at 98°C. For Western blot, 20 ug of protein lysates, plus 1x laemmli buffer and 50 mM DTT, were boiled for 5 min at 98°C. SDS-PAGE was run using 4–20% Mini-PROTEAN TGX precast gels (Bio-Rad, 4568094) and run at 130V for 1 hrs. Blotting was done using the Trans-Blot Turbo Transfer System (Bio-Rad). Blots were all blocked in 5% milk in TBST (TBS buffer, 0.1% tween detergent). Primary antibodies were incubated overnight at 4°C shaking. The following day, they were rinsed 3 times 10 min in TBST buffer, incubated with secondary antibody in TBST buffer for 1 hr at room temperature and rinsed 3 times 10 min in TBST buffer. Western imaging was done using ECL Western blotting substrate (Thermo Scientific, 32106) supplemented occasionally with SuperSignal West Femto Maximum Sensitivity Substrate (Thermo Scientific, 34095). Blots were exposed in a Bio-Rad Gel Doc XR system. Quantification of Western Blot was performed using Bio-Rad Image Lab software. Primary antibodies used: DDX3X (Sigma, HPA001648, 1:1000), FLAG-HRP (Sigma, A8592, 1:5000), RFP (Rockland, 600-401-379, 1:500).

## Fluorescence recovery after photobleaching (FRAP)

For FRAP of the live cells, cells expressing EGFP-HsDDX3X were cultured in 24well plate glass-bottomed dishes (Mattek, P24G-1.5-10-F). The FRAP assays were conducted using the bleaching module of the Zeiss LSM 780 confocal microscope using 63x magnification. The 488 nm laser was used to bleach the EGFP signal. Bleaching was focused on a circular region of interest (ROI) using 100% laser power. Time lapse images were collected every 30 sec starting 90sec before the bleaching for a total time of 20 min. A same-sized circular area away from the bleaching point was selected as an unbleached control. The fluorescence intensity was measured in Fiji.

## Quantification and statistical analysis

Experimenters were blinded at the analysis stage and all analyses were performed blindly by 1 or more investigators. Sample sizes were based on previous experiments [9,34,38]. No data was excluded from analysis. Number of data points and statistical tests used for all the comparisons are indicated in the figure legends.

## Supporting information

**S1 Table. Summary of clinical data for mutations examined in this study.**
(XLSX)

**S2 Table. Transcriptomic data and Gene Ontology analysis.**
(XLSX)

**S3 Table. Raw proteomic analysis.**
(XLSB)

**S4 Table. Normalized proteomic analysis.**
(XLSB)

**S1 Supportive Information. Source data with graph values.** Data used to generate graphs in all figures.
(XLSX)

**S1 Data. Source data for Western blot Fig 1. Full-length western blots for Fig 1. Note some blots were cut prior to probing.**
(TIF)

**S2 Data. Source data for Western blots Fig 5. Full-length western blots for Fig 5. Note some blots were cut prior to probing.**
(TIF)

**S1 Fig. DDX3X lentiviral expression transduction.** A) Low magnification representative images of mouse primary cultures 2 days after GFP-*HsDDX3X* lentiviral delivery. B) Quantification of transduction efficiency in A. Each dot represents a pooled primary culture. C) Snapshots of an unsuccessful mitosis and subsequent cell death. D) Quantification of failed divisions. E) Quantification of mitosis duration. F) Schematic example of a re-divisions. G) Quantifications of re-divisions. D, E, G) $\chi^2$ analysis with post-hoc Bonferroni. n = 3 live-imaging sessions and n = 3 litters. Scale bars: 10 μm. ****$p < 0.0001$; ns, not significant. Data are mean ± SD. "S": severe; "M": mild.
(TIF)

**S2 Fig. *Ddx3x* is not upregulated in DDX3X^WT or missense mutations conditions.** A, B) *Ddx3x* (A) and *Ddx3y* (B) levels from transcriptome datasets. C) Percentage of mapped reads from chromosome X and Y across all conditions and the three biological replicates of the transcriptome dataset. D) Gene ontology category relative to heatmap in Fig 3A. E) Quantification of sub cellular localization of S9.6+ cells. Two-way ANOVA. *$p < 0.01$; ns, not significant. Data are mean ± SD. "S": severe; "M": mild.
(TIF)

**S3 Fig. Turbo-ID Gene Ontology analysis highlights shared putative interactors in DDX3X^WT and missense mutations involved in RNA metabolism.** A) StringDB analysis of the 211 putative interactors shared between all experimental conditions. Nodes are divided according to GO categories. B, C, D, E) GO analysis of the significant interactors for R326H (B), R376C (C), R488H (D) and T532M (E) relative to the negative control.
(TIF)

**S4 Fig. DDX3X overexpression in N2A cells affects sub-cellular localization of protein interactors.** A–E) Gene Ontology analysis of the significant interactors for WT DDX3X (A). and those not shared with DDX3X^WT for R326H (B), R376C (C), T532M (D). E, F) Representative images of N2A cells transfected with GFP-HsDDX3X (green) and mScarlet-CFDP1 (red) (E)

and quantifications (F). G, H) N2A cells transfected with GFP-HsDDX3X (green) and mScarlet-SMURF2 (red) (G) and quantifications (H). Arrowheads point to example of whole cell localization, arrows to cytoplasmic localization. I, J) N2A cells transfected with GFP-HsDDX3X (green) and mScarlet-HOOK2 (red) (I) and quantifications (J). K, L) N2A cells transfected with GFP-HsDDX3X (green) and mScarlet-RHOB (red) (K) and quantifications (L). Arrowheads point to example of membrane localization, arrows to cytoplasmic localization. M, N) N2A cells transfected with GFP-HsDDX3X (green) and mScarlet-HSPBP1 (red) (M) and quantifications (N). F, H, J, L, N) each dot represents an independent transfection. Scale bars: 10 μm. Unpaired t-test $*p < 0.01$; $**p < 0.001$; $***p < 0.0001$; $****p < 0.0001$. Data are mean ± SD. (TIF)

**S5 Fig. GFP-HsDDX3X granules are present in progenitors and neurons overlap with endogenous DDX3X.** A) Quantification of total cell composition (SOX2 and Tuj1+ cells) of primary cells after 2 days of lentiviral transduction. B) Example of staining with anti-DDX3X (red) and GFP (green). Inserts show overlap with GFP-HsDDX3X granules. C) Example of staining with anti-DDX3X (red) and GFP (green). Inserts show overlap in nuclear puncta of DDX3X. Scale bars: 10 μm. Two-way ANOVA ns, not significant. Data are mean ± SD. "S": severe; "M": mild. (TIF)

## Acknowledgments

We thank the *DDX3X* Foundation and members of the Silver lab and Kate Meyer, Stephen Floor, Cagla Eroglu for helpful discussions and reading the manuscript. We thank Jianhong Ou for assistance with bioinformatics analysis, Carly Newman for mouse husbandry, Kate Meyer for plasmids. We thank the Duke proteomics, Regeneromics, and Microscopy core facilities.

## Author contributions

**Conceptualization:** Federica Mosti, Mariah L. Hoye, Debra L. Silver.

**Formal analysis:** Federica Mosti, Mariah L. Hoye, Carla F. Escobar-Tomlienovich.

**Funding acquisition:** Federica Mosti, Mariah L. Hoye, Debra L. Silver.

**Investigation:** Federica Mosti, Mariah L. Hoye.

**Methodology:** Federica Mosti, Mariah L. Hoye, Debra L. Silver.

**Project administration:** Debra L. Silver.

**Supervision:** Debra L. Silver.

**Validation:** Federica Mosti, Carla F. Escobar-Tomlienovich.

**Visualization:** Federica Mosti.

**Writing – original draft:** Federica Mosti, Debra L. Silver.

**Writing – review & editing:** Federica Mosti, Mariah L. Hoye, Carla F. Escobar-Tomlienovich, Debra L. Silver.

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
