## [Decision Letter · Decision Letter 0]

10 Dec 2024

PGENETICS-D-24-01170

Multi-modal investigation reveals pathogenic features of diverse DDX3X missense mutations

PLOS Genetics

Dear Dr. Silver,

Thank you for submitting your manuscript to PLOS Genetics. After careful consideration, we feel that it has merit but does not fully meet PLOS Genetics's publication criteria as it currently stands. Therefore, we invite you to submit a revised version of the manuscript that addresses the points raised during the review process.

Please submit your revised manuscript within 30 days Jan 09 2025 11:59PM. If you will need more time than this to complete your revisions, please reply to this message or contact the journal office at plosgenetics@plos.org. Please include the following items when submitting your revised manuscript:

We look forward to receiving your revised manuscript.

Kind regards,

Frank L Conlon

Academic Editor

PLOS Genetics

Scott Williams

Section Editor

PLOS Genetics

Aimée Dudley

Editor-in-Chief

PLOS Genetics

Anne Goriely

Editor-in-Chief

PLOS Genetics

**Additional Editor Comments:**

Please address the minor comments by the Reviewers. We look forward to receiving your edited version.

**Journal Requirements:**

At this stage, the following Authors/Authors require contributions: Federica Mosti, Mariah Hoye, Carla Escobar-Tomlienovich, and Debra L. Silver. Please ensure that the full contributions of each author are acknowledged in the "Add/Edit/Remove Authors" section of our submission form.

The list of CRediT author contributions may be found here: https://journals.plos.org/plosgenetics/s/authorship#loc-author-contributions

- ® on page: 27.

- TM on page: 27.

5) We notice that your supplementary Figures are included in the manuscript file. Please remove them and upload them with the file type 'Supporting Information'. Please ensure that each Supporting Information file has a legend listed in the manuscript after the references list.

Potential Copyright Issues:

- Figures: 1A, and 7E; Please confirm whether you drew the images / clip-art within the figure panels by hand. If you did not draw the images, please provide a link to the source of the images or icons and their license / terms of use; or written permission from the copyright holder to publish the images or icons under our CC BY 4.0 license. Alternatively, you may replace the images with open source alternatives. See these open source resources you may use to replace images / clip-art:

- https://openclipart.org/ .

7) We note that your Data Availability Statement is currently as follows: "All transcriptomic data can be accessed at: https://www.ncbi.nlm.nih.gov/geo/query/acc.cgi?acc=GSE279078 and by entering the following token: ehedgmuofjubjkj,All proteomics data (raw and normalized) are included in Figure 5 and Tables S3 and S4. All other data are available from the corresponding author upon reasonable request.". Please confirm at this time whether or not your submission contains all raw data required to replicate the results of your study. Authors must share the “minimal data set” for their submission. PLOS defines the minimal data set to consist of the data required to replicate all study findings reported in the article, as well as related metadata and methods (https://journals.plos.org/plosone/s/data-availability#loc-minimal-data-set-definition ).

- The points extracted from images for analysis..

If your submission does not contain these data, please either upload them as Supporting Information files or deposit them to a stable, public repository and provide us with the relevant URLs, DOIs, or accession numbers. For a list of recommended repositories, please see https://journals.plos.org/plosone/s/recommended-repositories .

8) Please ensure that the funders and grant numbers match between the Financial Disclosure field and the Funding Information tab in your submission form. Note that the funders must be provided in the same order in both places as well.

**Reviewers' comments:**

Reviewer's Responses to Questions

**Comments to the Authors:**

Reviewer #1: The manuscript by Mosti et al. entitled “Multi-modal investigation reveals pathogenic features of diverse DDX3X missense mutations” characterized four clinical missense mutations in DDX3X that cause mild to severe anatomical defects and behavioral phenotypes. Authors leveraged a primary mouse neural progenitor cell model to study the impact by these mutations on cell fate during early development. They first optimized the expression level of GFP-tagged human DDX3X in mouse, with the missense mutant proteins expressed at ~30% of the endogenous mouse DDX3X level. Under these conditions authors showed that these mutations differentially impact cell fate and survival, with the clinically severe mutations cause more proliferation, less neurogenic cell division, and more neuronal apoptosis. They also showed transcriptomic upregulation of DNA damage response pathways in cells expressing the mutants. They then investigated DNA damage levels by immuno-staining for DNA double strand breaks and R-loop. They showed high levels of gH2A.X and cytoplasmic R-loop associated with the severe mutations in Tuj1+ neurons, leading to the hypothesis that R-loop-induced DNA damage underlies neuronal apoptosis. Finally, they analyzed the protein interactome of DDX3X mutants as well as stress granule formation in cell expressing the mutants. This is the first study to demonstrate that DDX3X missense mutations impair neurogenesis and neuronal survival. The discovery of increased DNA damage and R-loop formation in cell expressing mutant DDX3X, and particularly those clinically severe mutants, bears importance on the etiological basis for DDX3X Syndrome and associated neurodevelopmental disorders. The experimental model is well chosen, the data are clearly presented, and the manuscript is well written.

I only have one major point and a couple of minor concerns/suggestions for the authors. The major point regards the observation of increased R-loop level mainly in the cytoplasm of cells expressing the mutants. The authors hypothesized that (nuclear) R-loops translocate into the cytoplasm, causing accumulation of stress granules, ultimately leading to apoptosis. This translocation hypothesis remains untested and may not be the parsimonious explanation. Because the S9.6 antibody is known to react with dsRNA in addition to the RNA:DNA hybrid, cytoplasmic S9.6 signals are largely attributable to dsRNA. Meanwhile, the nuclear S9.6 signals were not quantified. Based on the legend of Fig. 4, authors quantified the percentage of cells showing positive signals for S9.6 among GFP+ cells, instead of quantifying the level of S9.6 (or other DNA damage markers) signals. This practice might overlook those S9.6 signals in the nucleus, which are more likely due to co-transcriptional R-loop formation. Therefore, I suggest that the authors re-quantify nuclear S9.6 signals using Hoechst dye to define the nuclear perimeter and ask if mutant DDX3X cause increased nuclear R-loop formation as well. If feasible, it would be great to repeat the immunostaining for R-loop with and without RNase III treatment to remove dsRNA and then analyze nuclear S9.6 signals and stress granule formation in the cytoplasm. I feel these experiments would be crucial for demonstrating co-transcriptional R-loop formation causing DNA double strand breaks due to DDX3X mutations.

Minor points:

1. The statement “Due to its location on the X chromosome, DDX3X syndrome predominantly affects females, although a growing number of males have been identified” is a bit mis-leading. Typically an X-linked mutation would be considered to affect males predominantly instead. I thought the reason why DDX3X mutations show a female-bias is because hemizygous mutations in males are likely lethal.

2. The DNA damage response pathway markers ATM, ATR, DNA-PK, etc. all showed up-regulation in cells expressing the mutant DDX3X. These markers suggest that increased DNA damage might be due to DNA replication (ATR, DNA-PK) defects, in addition to transcription-associated DNA damage. This might be an important point given the observation of increased proliferation in the mutant cells.

3. Related to the point #2, might it be that the DNA damage in neurons is due to re-entry of the cell cycle? It would be nice to see some in-depth discussion on the specific DNA damage response.

Reviewer #2: This was a well written paper addressing an important question of the consequences of patient missense mutations in DDX3X. They used an NPC model system, and evaluated the consequences of 4 mutations in a very careful way, with a range of different approaches. They carefully thought about how to titrate the lenti to express the DDX3X variants and near endogenous levels. They confirmed their level of overexpression had no transcriptional effects. I like the use of the S and the M in the figures to keep track of which was severe and which was mild. They had nice follow up of initial transcriptional discoveries with cell biological approaches to support their conclusions. On the whole, it was an important and well executed paper that should merit publication in Plos Genetics, and they mostly support the claims they made. I have only a few minor points where I thought they needed a bit more support, and a few minor clarifying points.

Claims:

1) Diverse DDX3X missense mutations disrturb neural dev. through distinct cellular and molecular mechanisms, comparing two missense from severe patients and two from mild

2) Severe mutations impair neurogenesis and cause cell death, while mild mutations have a mild effect.

3) Different mutations have different interactors by proximity labeling, with severe ones sharing more neighbors.

4) Transcriptomics identifies pathways, including DNA damage response, and clinically severe mutations show DNA damage in neurons.

5) These may be due to DNA:RNA hybrids and formation of stress granules, pointing to a new mechanism of disease.

I found they mostly supported the claims, though it was not the clear to me the stress granules were part of the disease pathway (they could be a correlate), and I though they could strengthen the analysis for #3, as described below.

Suggestions:

A. It is a bit strange that DDX3X is having such a nuclear consequence for a cytoplasmic protein, and this doesn’t readily fit into their model. Could the assumption that DDX3X is cytoplasmic (as mentioned in the intro) be an oversimplification, and it is also in the nucleus at low levels or at certain puncta? The authors could consider some careful Immunofluorescent quantification of DDX3X (both their tagged versions, as well as endogenous) in the nucleus, to see if there are trace amounts. This may be possible from reanalysis of their existing collection of images. Or potentially this could be done with nuclear vs. cytoplasmic fractionation followed by Western, or perhaps over-ambitiously, Cut N Run to see if DDX3X associates with parts of DNA where R-loops are formed. Potentially their proteomics might tell them something too (is it associating with nuclear or cytoplasmic proteins). Overall, I recommend some limited additional experimentation exploring if its sub-cellular localization can support (or at least not refute) their model. If DDX3X does not colocalize with the Rloops staining, DSB, etc, in nucleus, this suggests that the impact of DDX3X on these phenomena is indirect rather than direct (perhaps disrupting translation of nuclear helicases or BRAC1A, for example).

B. In the proteomics, addition to comparing each tagged protein to untagged Turbo ID, if they want to conclude that mutants are different from WT DDX3X, they should also compare them directly and statistically to WT DDX3X. Of course, they should probably limit this analysis to the those proteins are significantly enriched in the first analysis of comparing the DDX3Xs untagged Turbo ID. I would recommend this as additional columns in existing sup sheets, as well as perhaps an additional figure panel like 5B, but with the log FC and p-value coded as WT vs Mutant. I think this added analysis is important because their primary conclusions for claim #3 are about the difference between WT and mutant, and thus their should be a statistical analysis that directly asks about the difference between WT and mutants. Currently it appears the analysis compares each of these to a third, untagged, samples instead. Such a direct comparison would improve their support of claim #3.

C. Do the authors have any samples from their historical DDX3X LOF mutants to look at marks of DNA damage and R loop formation? That would increase the impact of the pathways discovered here by showing some shared mechanism with the LOF perhaps. The negative results is also interesting, and would argue strongly LOF and severe missense are different diseases.

Finally, it is not necessary to support their claims, but it might be a cool experiment that further supports their model to overexpress RNASE H, or use some other treatment that resolves R loops and determine if this improves their severe mutants in terms of DSBs and apoptosis (e.g, https://www.nature.com/articles/s41467-020-18306-x ), or cytoplasmic R-loops specifically, if they think these are mediating the apoptosis, as was shown in this recent study: https://www.nature.com/articles/s41586-022-05545-9 . Likewise, do the authors think it is the stress granules that matter for death, or activation of IRF3 signaling as in https://www.nature.com/articles/s41586-022-05545-9? This could be at least worth some discussion, if not some Westerns similar to the prior work.

Minor:

In figure 8, is the little purple-blue shape meant to be DDX3X? Maybe add that to the key.

Methods: how the statistical comparison of samples was done for proteomics is missing.

Supplemental tables could each use legends or README sheets to explain what the column headers mean. This will make the data reusable for others.

How do they think about X inactivation. Is that relevant here? Or does this escape X-inactivation? If so, it might be worth mentioning, at least parenthetically.

2F,G. What are the two factors in the two way anova? I only see one factor, unless sex was the other? I have a similar question about other figures.

DDX3Y western, to see if it is upregulated at protein level, would also be interesting (though could be tangential, and not required). But, perhaps they could at least discuss why increased DDX3Y does not rescue DDX3X deficiency?

Line 254: for a second I thought they might have had a typo to mean ‘nuclear accumulation of R loops’ rather than cytoplasmic? So it was initially a bit confusing . T532M looks nuclear, and R326H looks cytoplasmic. But, that is because I was not aware of the very recent literature on cytoplasmic R loops. It might be worth mentioning those citations right away, since cytoplasmic R loops are a recently reported phenomena.

Reviewer #3: In this study, the Authors characterise the cellular and molecular mechanisms underlying clinically severe and mild mutations by integrating transcriptomics, proteomics, and live imaging. The study uses primary mouse neural progenitors to reveal how these mutations differentially affect neurogenesis, neuronal survival, and key pathways such as RNA metabolism and DNA damage response.

Suggested Additions:

1. Discussion Expansion:

The discussion should delve deeper into the role of R-loops in inducing neuroinflammation. Highlight their capacity to trigger immune responses, emphasizing the link between unresolved DNA:RNA hybrids and chronic inflammation in neural tissues. This could include potential therapeutic approaches targeting R-loop resolution to mitigate inflammation and its downstream effects.

2. Conclusion Expansion:

The conclusion should outline how these findings can inform clinical strategies, particularly the development of targeted therapies for severe DDX3X mutations. Highlight opportunities for personalised medicine based on molecular phenotyping of specific mutations.

**Have all data underlying the figures and results presented in the manuscript been provided?**

Reviewer #1: Yes

Reviewer #2: Yes

Reviewer #3: Yes

PLOS authors have the option to publish the peer review history of their article (what does this mean? ). If published, this will include your full peer review and any attached files.

**Do you want your identity to be public for this peer review?** For information about this choice, including consent withdrawal, please see our Privacy Policy .

Reviewer #1: No

Reviewer #2: No

Reviewer #3: **Yes: ** Domenico Plantone

**Figure resubmission:**

While revising your submission, please upload your figure files to the Preflight Analysis and Conversion Engine (PACE) digital diagnostic tool, https://pacev2.apexcovantage.com/ . PACE helps ensure that figures meet PLOS requirements. To use PACE, you must first register as a user. Registration is free. Then, login and navigate to the UPLOAD tab, where you will find detailed instructions on how to use the tool. If you encounter any issues or have any questions when using PACE, please email PLOS at figures@plos.org. Please note that Supporting Information files do not need this step. If there are other versions of figure files still present in your submission file inventory at resubmission, please replace them with the PACE-processed versions.
---

## [Editor Report · Decision Letter 1]

27 Dec 2024

Dear Dr Silver,

We are pleased to inform you that your manuscript entitled "Multi-modal investigation reveals pathogenic features of diverse DDX3X missense mutations" has been editorially accepted for publication in PLOS Genetics. Congratulations!

Yours sincerely,

Frank L Conlon

Academic Editor

PLOS Genetics

Hua Tang

Section Editor

PLOS Genetics

Aimée Dudley

Editor-in-Chief

PLOS Genetics

Anne Goriely

Editor-in-Chief

PLOS Genetics

Comments from the reviewers (if applicable):

**Data Deposition**

http://datadryad.org/submit?journalID=pgenetics&manu=PGENETICS-D-24-01170R1

More information about depositing data in Dryad is available at http://www.datadryad.org/depositing . If you experience any difficulties in submitting your data, please contact help@datadryad.org for support.

**Press Queries**

---

## [Editor Report · Acceptance letter]

PGENETICS-D-24-01170R1

Multi-modal investigation reveals pathogenic features of diverse DDX3X missense mutations

Dear Dr Silver,

We are pleased to inform you that your manuscript entitled "Multi-modal investigation reveals pathogenic features of diverse DDX3X missense mutations" has been formally accepted for publication in PLOS Genetics! Your manuscript is now with our production department and you will be notified of the publication date in due course.

With kind regards,

Anita Estes

PLOS Genetics

On behalf of:
